# Accessing CKM suppressed top decays at the LHC

Darius A. Faroughy*
*Physik-Institut, Universität Zürich, CH-8057, Switzerland*

Jernej F. Kamenik†
*Jožef Stefan Institute, Jamova 39, 1000 Ljubljana, Slovenia and*
*Faculty of Mathematics and Physics, University of Ljubljana, Jadranska 19, 1000 Ljubljana, Slovenia*

Manuel Szewc‡
*Jožef Stefan Institute, Jamova 39, 1000 Ljubljana, Slovenia*

Jure Zupan§
*Department of Physics, University of Cincinnati, Cincinnati, Ohio 45221, USA*

We propose an extension of the existing experimental strategy for measuring branching fractions of top quark decays, targeting specifically $t \to j_q W$, where $j_q$ is a light quark jet. The improved strategy uses orthogonal $b$- and $q$-taggers, and adds a new observable, the number of light-quark-tagged jets, to the already commonly used observable, the fraction of $b$-tagged jets in an event. Careful inclusion of the additional complementary observable significantly increases the expected statistical power of the analysis, with the possibility of excluding $|V_{tb}| = 1$ at 95% C.L. at the HL-LHC, and accessing directly the standard model value of $|V_{td}|^2 + |V_{ts}|^2$.

## I. INTRODUCTION

The $V_{tx}$ elements of the third row of the CKM matrix are currently well constrained only indirectly, from measurements of radiative $B$ meson decays and neutral $B_{s,d}$ meson oscillations which involve loops with virtual top-quarks. A recent global CKM fit gives [1] (see also [2])

$$
\begin{aligned}
|V_{tb}^{\rm SM}| &= 999.118^{+0.031}_{-0.036} \times 10^{-3}\,, \\
|V_{ts}^{\rm SM}| &= 41.10^{+0.83}_{-0.72} \times 10^{-3}\,, \\
|V_{td}^{\rm SM}| &= 8.57^{+0.2}_{-0.18} \times 10^{-3}\,.
\end{aligned}
\tag{1}
$$

These can be compared with direct measurements of $|V_{tx}|$, from productions of on-shell top quarks at the LHC and their decays. The measurements of $b$-jet fractions in $t \to Wj$ top decays currently set a bound [3]

$$
\mathcal{R}_b \equiv \frac{\mathcal{B}(t \to bW)}{\sum_{j=d,s,b} \mathcal{B}(t \to jW)} > 0.955 \ @ \ 95\% \ \text{C.L.}\,,
\tag{2}
$$

which can be interpreted as $\sqrt{|V_{td}|^2 + |V_{ts}|^2} < 0.217|V_{tb}|$. A less precise direct measurement of the $|V_{ts}|$ and $|V_{td}|$ matrix elements was performed in Ref. [4], using $t$-channel single top production. Alternative ways of directly measuring $|V_{tx}|$ were also proposed, either using $tW$ associated production [5], or by $s$-tagging top-quark decay products [6]. All of these approaches suffer from low statistics due to the smallness of $\sqrt{|V_{td}|^2 + |V_{ts}|^2}$ and are thus not expected to match the precision of the SM prediction from the CKM global fits, Eq. (1). The situation is very different for the matrix elements in the first two rows of the CKM matrix, which are already probed directly with ever improving precision using decays of nuclei, kaons, charmed mesons and $B$-hadrons [7–11]. The main goal of the present manuscript is to advance the tools for such direct measurements also for the $|V_{tx}|$ CKM elements.

* faroughy@physik.uzh.ch
† jernej.kamenik@cern.ch
‡ manuel.szewc@ijs.si
§ zupanje@ucmail.uc.edu

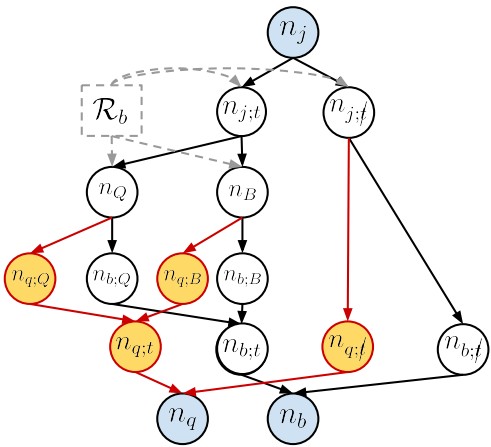

**Figure 1:** Illustration of the probabilistic model for determining $\mathcal{R}_b$. For a given $pp \to t\bar{t}$ leptonic channel, with $\ell\ell'$ lepton flavors, the events are distributed into bins with $n_j$ jets. The events are split into bins with $n_q$ light-tagged and $n_b$ $b$-tagged bins. The arrow shows the probabilities for each event with $n_j$ jets to end up in the $\{n_b, n_q\}$ bin. The measured $n_j, n_b, n_q$ (dark blue circles) can be related through latent, marginalized over, observables. We show as yellow filled circles and red arrows the added variables we propose to extend the probabilistic model proposed in Ref. [3] with, and increase the statistical power of the analysis.

The proposed novel analysis strategy to measure $\mathcal{R}_b$ builds upon Ref. [3] and targets the dileptonic $t\bar{t}$ signal region at the LHC. By applying a set of orthogonal $b$- and $q$-taggers to the final state jets in these events, one can go beyond determining $\mathcal{R}_b$ from fractions of $b$-tagged jets. In particular, by carefully analyzing multiplicity distributions of both $b$-quark and $q$-quark jets produced in top-quark decays and taking advantage of the complementarity between the measured numbers of $b$-tagged jets, $n_b$, and $q$-tagged jets, $n_q$, we are able to significantly improve the expected precision in the measurement of $\mathcal{R}_b$. We project that this could allow the (HL-)LHC to establish $\mathcal{R}_b < 1$ in the SM despite the low statistics of $t \to j_q W$ due to the smallness of $\sqrt{|V_{td}|^2 + |V_{ts}|^2}$.

The paper is organized as follows. In Section II we review the probabilistic model of Ref. [3] and modify it to incorporate $n_q$. In Section III, we validate and evaluate the model on two pseudo-datasets obtained from simulations with different $\mathcal{R}_b$ values. In Section IV we propose an analysis strategy that combines the state of the art $b$-taggers with an orthogonal $q$-tagger and obtain expected results for different LHC center-of-mass energies and luminosities. In Section V we summarize the main results and discuss possible improvements of the presented analysis.

## II.   PROBABILISTIC MODEL

We start by introducing the probabilistic model that can be used for measuring $\mathcal{R}_b$ from a $pp \to t\bar{t}$ dataset, where both tops decay leptonically, either through $t \to bW$ or $t \to qW$, followed by $W \to \ell\bar{\nu}$, resulting in a dilepton final state. It is an extension of the model used in Ref. [3], where we include also the dependence on the number of light quark tagged jets, $n_q$, see Fig. 1. Correspondingly, the model can be reduced to the one of Ref. [3] simply by marginalizing over the new variables (shown in yellow circles in Fig. 1). In the remainder of this section we describe the likelihood analysis that compares measurements with the expected event yields from the probabilistic model, and then provide the test statistics sensitive to $\mathcal{R}_b$.

The $pp \to t\bar{t}$ events (including background events) are split into different categories, labelled by $\{\ell\ell', n_j\}$, and then further divided into $\{n_b, n_q\}$ bins. Here $\ell\ell'$ are the flavors of the two final state leptons, $\ell\ell' = e^+e^-, \mu^+\mu^-, e^\pm\mu^\mp$, while $n_j = 2, 3, 4$, is the number of jets in the event, among which $n_b$ are tagged as $b$-jets, and $n_q$ as light-quark jets, with $n_j \geq n_b + n_q$. The expected number of events $\bar{N}_{\ell\ell'}$ in the $\{\ell\ell', n_j, n_b, n_q\}$

bin is given by

$$\bar{N}_{\ell\ell'}(n_b, n_q | n_j) = P_{\ell\ell'}(n_b, n_q | n_j, \mathcal{R}_b, \theta_i) N_{\ell\ell'}(n_j), \tag{3}$$

where $N_{\ell\ell'}(n_j)$ is the number of observed events that have $n_j$ jets,

$$N_{\ell\ell'}(n_j) = \sum_{n_b, n_q} N_{\ell\ell'}(n_b, n_q | n_j). \tag{4}$$

The probability $P_{\ell\ell'}(n_b, n_q | n_j, \mathcal{R}_b, \theta_i)$ of observing $n_b$ b-tagged and $n_q$ q-tagged jets depends on $\mathcal{R}_b$, the parameter we are interested in, as well as on a number of nuisance parameters, $\theta_i$, discussed below. Comparing the expected number of events, $\bar{N}_{\ell\ell'}(n_b, n_q | n_j)$, with the observed number of events in the $\{\ell\ell', n_j, n_b, n_q\}$ bin, $N_{\ell\ell'}(n_b, n_q | n_j)$, one can then measure $\mathcal{R}_b$, if the $P_{\ell\ell'}$ dependence on $\mathcal{R}_b$ is known. The main purpose of this manuscript is to show how to build a probabilistic model for $P_{\ell\ell'}(n_b, n_q | n_j, \mathcal{R}_b, \theta_i)$ using data and to show that the high-luminosity LHC $pp \to t\bar{t}$ data is expected to have already nontrivial sensitivity to the SM value of $\mathcal{R}_b$. The schematic of the probabilistic model, in terms of the variables already used in Ref. [3] (solid white circles) as well as the new variables introduced here (yellow filled circles), is shown in Fig. 1. The form of $P_{\ell\ell'}(n_b, n_q | n_j, \mathcal{R}_b, \theta_i)$, including the explicit dependence on $n_b, n_q, n_j, \mathcal{R}_b$, and $\theta_i$, is given in App. A.

To measure $\mathcal{R}_b$ with the probabilistic model, we construct a log likelihood

$$\mathcal{L}(\mathcal{R}_b, \theta_i) = \prod_{\ell\ell'} \prod_{n_j=2}^{4} \prod_{n_b=0}^{n_j} \prod_{n_q=0}^{n_j-n_b} \mathcal{P}(N_{\ell\ell'} | \bar{N}_{\ell\ell'}) \prod_i \rho(\theta_i), \tag{5}$$

where we shortened $N_{\ell\ell'}(n_b, n_q | n_j)$ and $\bar{N}_{\ell\ell'}(n_b, n_q | n_j)$ to just $N_{\ell\ell'}, \bar{N}_{\ell\ell'}$ for clarity, $\mathcal{P}$ is the Poisson probability density function and $\rho$ is a probability distribution to be discussed below. We consider the following nuisance parameters $\theta_i$,

- $f_{t\bar{t}}$: the fraction of $pp \to t\bar{t}$ events out of all the observed events, including the background. There is one $f_{t\bar{t}}$ for each $\{\ell\ell', n_j\}$ category, i.e., there are nine separate $f_{t\bar{t}}$ (here and below we suppress the category labels in order to shorten the notation). As detailed in App. A, we obtain $f_{t\bar{t}}$ with a maximum likelihood fit per $\{\ell\ell', n_j\}$ category with no further information.

- $k_{\mathrm{st}}$: the fraction of single-top events relative to the fitted amount of $t\bar{t}$ events. There are nine $k_{\mathrm{st}}$ parameters, one for each $\{\ell\ell', n_j\}$ category, which are derived from the fitted $f_{t\bar{t}}$ with the procedure given in App. A.

- $f_{\ell j; t}$: is the fraction of jets originating from a top decay out of all measured jets in the sample. There are nine $f_{\ell j; t}$ parameters, one for each $\{\ell\ell', n_j\}$ category, and they are fitted from the invariant mass spectrum of all lepton-jet pairs as detailed in App. A.

- $\epsilon_\beta^\alpha$: are the jet tagging efficiencies with the upper labels denoting different taggers, $\alpha = b, q, g$, for b-jet, q-jet, and gluon jet taggers, respectively. The lower indices are denoting the flavor and the origin of the true hard object, $\beta = \{B, Q, j; \bar{t}\}$, where $\beta = B, Q$, for jets initiated by hard $b$ or hard $Q = d, s$ quarks coming from top decays, respectively, while $\beta = "j; \bar{t}"$ denotes that a jet was initiated by a hard parton from ISR/FSR or background processes, i.e., not from a top decay. The taggers need to be orthogonal to ensure that a given jet gets assigned a unique tag, $b$, $q$ or $g$. For each of the true objects the efficiencies also sum up to one,

$$\sum_\alpha \epsilon_\beta^\alpha = 1, \ \forall \beta. \tag{6}$$

For each $\beta = \{B, Q, j; \bar{t}\}$ only two tagging efficiencies are thus independent, and we take these to be $\epsilon_\beta^b$ and $\epsilon_\beta^q$. For $\beta = B, Q$ the tagging efficiencies are independent of the flavor of the leptonic final state and of the total number of jets, leading to four independent parameters, $\epsilon_{b,q}^{B,Q}$. These are estimated using auxiliary measurements with high-purity samples, see for example Ref. [12], where $\epsilon_\beta^b$ were determined using multijet and $t\bar{t}$ events. For $\beta = "j; \bar{t}"$ the tagging efficiency is in principle background/final state dependent, and thus we introduce two parameters $\epsilon_{j; \bar{t}}^{b,q}$ for each of the $\{\ell\ell', n_j\}$ categories, to be fitted along with $\mathcal{R}_b$.

The nuisance parameters (except $\epsilon_{j;\ell}^{b,q}$) are set to their central values $\theta_i^0$, determined by the auxiliary measurements as discussed above. For $f_{t\bar{t}}$, $k_{\mathrm{st}}$ and $f_{\ell j;t}$, the auxiliary measurements refer to the measured differential distributions where no $b$- or $q$-tagging is applied to the jets. Except for the subleading dependence of $k_{\mathrm{st}}$ these auxiliary measurements do not depend on $\mathcal{R}_b$. We then fit for possible deviations of $\theta_i$ from their central values $\theta_i^0$. Because all of the above nuisance parameters are normalization uncertainties, we parameterize the deviations following Ref. [13], and introduce additional variables $\eta_i$ such that

$$\theta_i = \theta_i^0 f(\eta_i; 1, I_i^+, I_i^-, 1). \tag{7}$$

Here $f(\eta_i; 1, I_i^+, I_i^-, 0)$ is the polynomial interpolation and exponential extrapolation defined in Ref. [13],

$$f(\eta_i; 1, \eta_i^+, \eta_i^-, 1) = \begin{cases} \left(I_i^+\right)^{\eta_i} & \text{if } \eta_i \geq 1\,, \\ 1 + \sum_{j=1}^{6} a_{ji}\eta_i^j & \text{if } |\eta_i| < 1\,, \\ \left(I_i^-\right)^{-\eta_i} & \text{if } \eta_i \leq -1\,, \end{cases} \tag{8}$$

with $I_i^+$, $I_i^-$ the $\pm 1\sigma$ variations of the nuisance parameter $\theta_i$, while the six coefficients $a_{ji}$ are fixed by demanding continuity of $f$ and its first two derivatives at $\eta_i = \pm 1$. The relevant $I^\pm$ can be obtained from the relative percentile uncertainties reported in Tables I and III. With the parameterization of $\theta_i$ in Eq. (7), the constraint $\rho(\theta_i)$ in Eq. (5) becomes

$$\rho(\theta_i) \to \mathcal{G}(0, \eta_i, 1). \tag{9}$$

Here, $\mathcal{G}$ is the Gaussian probability density function with mean $\eta_i$ and standard deviation 1, evaluated at 0. The form of $\rho(\theta_i)$ incorporates in an unbiased way the effect of systematic uncertainties, allowing for some variation of nuisance parameters $\theta_i$ around their central values. While in the maximization of the log-likelihood, Eq. (5), we use $\eta_i$ as the variables whose values are fit, we will continue to refer to $\theta_i$ as the nuisance parameters, but with the understanding, that the statistical analyses are always performed using the parameterization in Eq. (7).

To set confidence levels on $\mathcal{R}_b$ we follow the standard statistical techniques [14] and define the Profile Likelihood Ratio (PLR) $\lambda(\mathcal{R}_b)$

$$\lambda(\mathcal{R}_b) = \frac{\mathcal{L}(\mathcal{R}_b, \hat{\hat{\theta}}_i(\mathcal{R}_b))}{\mathcal{L}(\hat{\mathcal{R}}_b, \hat{\theta}_i)}, \tag{10}$$

and its associated test statistic

$$-2 \, \mathrm{Ln} \, \lambda(\mathcal{R}_b). \tag{11}$$

Here, $\hat{\hat{\theta}}_i(\mathcal{R}_b)$ are the maximum likelihood estimates of the nuisance parameters, obtained by maximizing $\mathcal{L}(\mathcal{R}_b, \theta_i)$, varying $\theta_i$, but keeping $\mathcal{R}_b$ fixed. The maximum likelihood estimates, $\hat{\mathcal{R}}_b$, $\hat{\theta}_i$, are then obtained by finding the global maximum of $\mathcal{L}(\mathcal{R}_b, \theta_i)$, varying both $\theta_i$ and $\mathcal{R}_b$.

We can also incorporate the constraint $\mathcal{R}_b \leq 1$, by modifying the PLR

$$\tilde{\lambda}(\mathcal{R}_b) = \begin{cases} \mathcal{L}(\mathcal{R}_b, \hat{\hat{\theta}}_i(\mathcal{R}_b))/\mathcal{L}(\hat{\mathcal{R}}_b, \hat{\theta}_i), & \text{if } \hat{\mathcal{R}}_b \leq 1\,, \\ \mathcal{L}(\mathcal{R}_b, \hat{\hat{\theta}}_i(\mathcal{R}_b))/\mathcal{L}(1, \hat{\hat{\theta}}_i(1)), & \text{if } \hat{\mathcal{R}}_b > 1\,. \end{cases} \tag{12}$$

with the associated test statistic

$$q = -2 \, \mathrm{Ln} \, \tilde{\lambda}(\mathcal{R}_b). \tag{13}$$

The test statistics $q$ is used in Section IV below to obtain the projected significance of rejecting at HL-LHC the $\mathcal{R}_b = 1$ hypothesis, i.e., the hypothesis that $|V_{td}|^2 + |V_{ts}|^2 = 0$, assuming true value of $\mathcal{R}_b$ is the SM one,

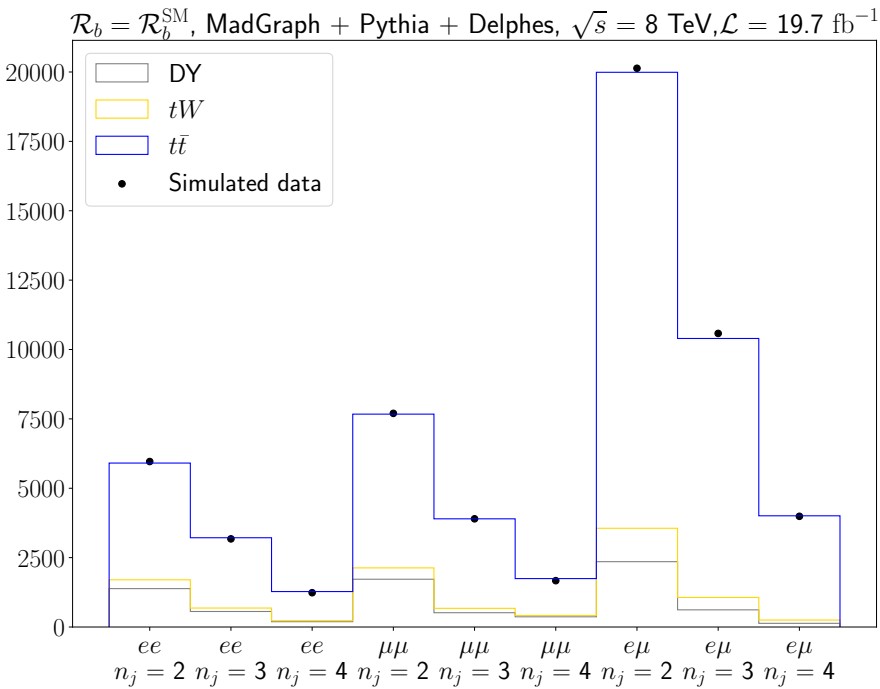

**Figure 2:** The expected number of events for $\mathcal{R}_b = \mathcal{R}_b^{\mathrm{SM}}$ in each of the $\{\ell\ell', n_j\}$ categories, for $t\bar{t}$ signal (blue), as well as the $tW$ (yellow) and Drell-Yan (green) backgrounds, showed in a stacked histogram form. An example of simulated data is denoted with black dots.

$\mathcal{R}_b = \mathcal{R}_b^{\mathrm{SM}}$. The expected median significance is then given by $\sqrt{q_1}$, where[1]

$$
q_1 = \begin{cases} -2 \, \mathrm{Ln} \, \lambda(1) & \text{if } \hat{\mathcal{R}}_b \leq 1 \,, \\ 0 & \text{if } \hat{\mathcal{R}}_b > 1 \,. \end{cases} \tag{14}
$$

The minimizations of test statistics following from Eqs. (10) and (12) were performed using the `iminuit` python package [15]. The quality of the fits depends on how well the statistical model can approximate data. This is a general problem; even if one incorporates many nuisance parameters, the fit will only be as good as the modelling assumptions. We know the model to be imperfect, since we are, for instance, ignoring the dependence of tagging efficiencies on jet $p_T$. We thus need to validate our model, similar to Ref. [3], by performing closure tests to verify whether the fit procedure is unbiased and the probabilistic model is a good approximation of the true probability density. To do this while also studying the benefits of adding $n_q$ to improve the fit, in Section III we explicitly generate two benchmark examples and perform the fits on the two examples. Having validated our model, in Section IV we detail our proposal for a direct measurement of $\mathcal{R}_b$ using data binned in $n_b$ and $n_q$ and study the expected performance by applying the generative model to the $N_{\ell\ell'}(n_j)$ values obtained from Monte Carlo simulations.

## III. MODEL EVALUATION

In this section, we perform fits to the maximum likelihoods in Eqs. (10) and (12) for two examples of pseudo-data. This both validates the use of probabilistic models, and gives an estimate of the improvement one can expect when including $n_q$ information in the fits.

---

[1] We do not impose explicitly the requirement $\mathcal{R}_b \geq 0$ since the data prefer large $\mathcal{R}_b \simeq 1$ value, and thus $\mathcal{R}_b$ never approaches the lower limit.

| $\alpha$-tagger | Cuts | $\epsilon_Q^\alpha$ | $\epsilon_B^\alpha$ |
|---|---|---|---|
| $b$-tagger | $N_{\mathrm{SV}} \geq 2$ | $0.002^{+10\%}_{-10\%}$ | $0.49^{+5\%}_{-5\%}$ |
| $q$-tagger | $N_{\mathrm{SV}} = 0 \ \& \ N_{\mathrm{const}} < 20$ | $0.69^{+20\%}_{-20\%}\%$ | $0.16^{+10\%}_{-10\%}$ |

**Table I:** Tagging and mis-tagging efficiencies for the $b$- and $q$-taggers, see main text for details.

First, we determine the expected number of events in each of the $\{\ell\ell', n_j, n_b, n_q\}$ bins using the `Madgraph` [16], `Pythia` [17], `Delphes` [18] simulation pipeline for $\sqrt{s} = 8\,\mathrm{TeV}$ LHC collision energy, and integrated luminosity $\mathcal{L} = 19.7\,\mathrm{fb}^{-1}$. We consider two benchmarks, $\mathcal{R}_b = \mathcal{R}_b^{\mathrm{SM}} \approx 0.998$ and $\mathcal{R}_b = 0.9$, and assume CKM unitarity, $\sum_{q=d,s,b} |V_{tq}|^2 = 1$. The Monte Carlo data contain events from the following production channels: $t\bar{t}$ with up to two additional jets, $tW$ with no additional jets, and Drell-Yan with up to two additional jets. For each of the two benchmarks we then construct an example of a possible experimental outcome – the pseudo-data. That is, for each of the $\{\ell\ell', n_j, n_b, n_q\}$ bins we sample a Poisson distribution with the average equal to the expected number of events in that bin, determined by the above Monte Carlo simulation. Fig. 2 shows the expected number of events in each $\{\ell\ell', n_j\}$ category for the $\mathcal{R}_b = \mathcal{R}_b^{\mathrm{SM}}$ benchmark. The contributions from $t\bar{t}$, $tW$, and Drell-Yan production are denoted with blue, yellow, and green, respectively. The black dots show an example of a generated pseudo-data, which, as anticipated, straddle the expected number of events in each category.

To perform the $\{n_b, n_q\}$ binning of the Monte Carlo data we implement simple orthogonal $b$- and $q$-taggers, by applying at `ROOT` [19, 20] level the cuts on secondary vertex multiplicity, $N_{\mathrm{SV}}$ [12], and constituent multiplicity of the jet, $N_{\mathrm{const}}$ [21], as listed in Table I. The secondary vertex multiplicity in the jet, $N_{\mathrm{SV}}$, is defined as the number of tracks within an angular distance $\Delta R \leq 0.3$ of the jet axis, with $p_T \geq 1$ GeV, and the transverse impact parameter $2.5 \ \mu\mathrm{m} \leq d_0 \leq 2.0$ mm. Here, $d_0$ is the transverse distance to the primary vertex at the point of closest approach in the transverse plane. The use of constituent multiplicity of the jet, $N_{\mathrm{const}}$, is motivated by its discriminating power between quarks and gluons [21]. While $N_{\mathrm{const}}$ is an IRC-unsafe observable, this poses no problems for our application, since we only require that it is a measurable property with discriminative power and do not intend to match it to perturbative calculations. While the use of just $N_{\mathrm{SV}}$ and $N_{\mathrm{const}}$ as discriminating observables leads to suboptimal taggers, this suffices for our purposes, i.e., demonstrating the usefulness of probabilistic modeling. Our analysis can be viewed as conservative, and one could improve on it in the actual experimental set-up by using better orthogonal taggers.

The choice of $N_{\mathrm{SV}}$ and $N_{\mathrm{const}}$ cuts, listed in Table I, is motivated by the $N_{\mathrm{SV}}$ and $N_{\mathrm{const}}$ distributions for jets with $p_T \geq 100$ GeV and $|\eta| \leq 2.4$ in the simulated $t\bar{t}$ sample, shown in Fig. 3. The jets were clustered using the anti-$k_T$ algorithm [22] with $R = 0.5$, and assigned a true flavor using the FlavorAlgorithm implemented in `Delphes`. This algorithm assigns a flavor to a jet by looking at the parton list remaining after showering and radiation and selecting the parton with no parton daughters that best explains its properties. Looking within a $\Delta R$ cone of the jet central axis, the algorithm labels as $b(c)$-quarks all the jets that contain a $b(c)$-quark parton and as $q$-quarks (gluons) those that do not and where the hardest parton is a light-quark (gluon). Because we are interested in tagging hard $q$-quarks and $b$-quarks originating from the top decays, the flavor definition implemented by `Delphes` is well suited for our purposes, i.e., to estimate $\epsilon_{B,Q}^\alpha$. On the other hand, the jets that do not originate from top quark decays cannot be properly matched to any single distribution shown in Fig. 3, which is why $\epsilon_{j;\not{t}}^\alpha$ are fitted along with $\mathcal{R}_b$ and the nuisance parameters for $\epsilon_{B,Q}^\alpha$. The jets with $N_{\mathrm{SV}} \geq 2$ are almost entirely due to an initial hard $b$-quarks (there are only very few $c$-quarks in the sample), cf. Fig. 3 (left). The jets with $N_{\mathrm{const}} \leq 20$, on the other hand, are more likely to be due to an initial hard $q$-quark than from a hard gluon (with almost no discriminating power between $b$- and $q$-quark initiated jets), cf. Fig. 3 (right).

The working point (WP) efficiencies, $\epsilon_\beta^\alpha$, for the $b$- and $q$-taggers in Table I, are determined from the Monte Carlo data as the fraction of $\beta$-quarks that is selected by the $\alpha$-tagger after applying the $N_{\mathrm{SV}}, N_{\mathrm{const}}$ cuts. Note that the $q$-tagger is a combination of an anti-$b$-tagger (the $N_{\mathrm{SV}}$ cut) and a quark/gluon-tagger (the $N_{\mathrm{const}}$ cut). This combination ensures orthogonality, i.e., that the $q$-tagged and $b$-tagged jets do not overlap. That the $q$-tagged sample of jets is obtained through a combined application of a quark/gluon tagger and an anti-$b$ tagger was then also used in writing the explicit expression for the probabilistic model, see Appendix A, and in particular relation (A27) which is needed to obtain the final expression, Eq. (A31). In Section IV,

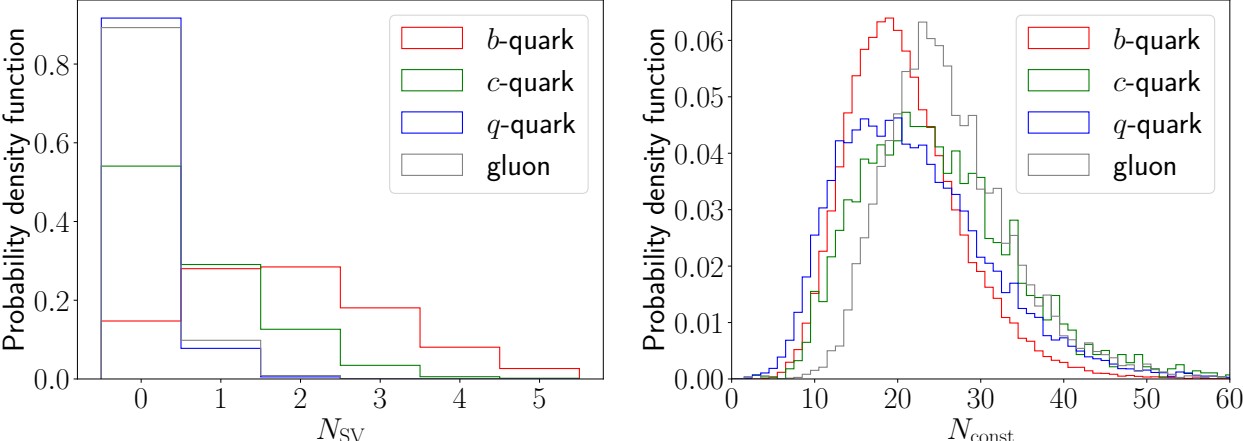

**Figure 3:** Normalized distribution of the discriminant variables $N_{\mathrm{SV}}$ (left) and $N_{\mathrm{const}}$ (right) for jets in a simulated $t\bar{t}$ sample, with either $b$-, $c$-, $q$- or gluon flavor assignments (see the main text for details on jet flavor assignments). The two variables are then used for the construction of simple $b$- and $q$-taggers.

we explicitly differentiate between the quark/gluon-tagger and the anti-$b$-tagger to incorporate state-of-the-art $b$-taggers. We also improve on the analysis performed in this section, by utilizing two working points for the $b$-tagger.

We use a relatively tight $b$-tagger WP to obtain a high sample purity (very low $\epsilon_Q^b$) at the price of losing many $b$-quarks (relatively low $\epsilon_B^b$). We expect this to be a well justified trade-off due to the high statistics of available dileptonic $t\bar{t}$ events. For the $q$-tagger, on the other hand, the high statistics is offset by the smallness of $V_{td,ts}$. We therefore select a medium WP which reduces the sample purity (medium $\epsilon_B^q$) but is able to capture more $q$-quarks (medium $\epsilon_Q^q$). We assume rather conservative systematic uncertainties on $\epsilon_\beta^\alpha$, a factor of several larger than those reported, e.g., in Refs. [12, 23] (in Table I the systematic uncertainties are listed as percentages of the central values). If systematics uncertainties were underestimated, this would exhibit itself through large absolute values of the fitted nuisance parameters (larger than about 2), when profiling over the log likelihoods in Eqs. (10) and (12). We do not find any such problems, and are thus lead to conclude that the systematic uncertainties quoted in Table I are large enough, and may even be lowered without encountering any tensions with the data.

The results of the fit to the pseudo-data are shown in Table II. The Negative Log Likelihood (NLL) is constructed either using $\{n_b, n_q\}$ bins, i.e., as in Eq. (11), or after summation over $n_q$, i.e., by using only binning in $\{n_b\}$. The first step in the fitting procedure is to determine for each $\{\ell\ell', n_j\}$ category the corresponding $f_{t\bar{t}}$, $k_{\mathrm{st}}$ and $f_{\ell j;t}$ (see App. A for details, for $\mathcal{R}_b \neq 1$ the extracted value of $k_{\mathrm{st}}$ is corrected according to Eq. (A3)). The extracted values of $f_{t\bar{t}}$, $k_{\mathrm{st}}$ and $f_{\ell j;t}$ are consistent with the results reported in Ref. [3], especially given that we take into account only the most relevant processes.

In the next step, $-2 \ln \lambda(\mathcal{R}_b)$ is minimized with respect to $\mathcal{R}_b$ and nuisance parameters using `iminuit`. The 95% C.L. intervals for the extracted value of $\hat{\mathcal{R}}_b$ are quoted in Table II. The results in the second row are obtained from a fit to pseudo-data binned in $\{n_b\}$ bins, and are consistent with the result reported in Ref. [3], $\hat{\mathcal{R}}_b = 1.014 \pm 0.003(\mathrm{stat}) \pm 0.032(\mathrm{syst})$. This is encouraging, and a welcome check of our set-up, especially given that we are not including the full set of systematic uncertainties. Using the probabilistic model one can extract appropriate confidence intervals and capture the essential physics. The results in the third row of Table II are obtained using pseudo-data in $\{n_b, n_q\}$ bins. We observe that this leads to significantly tighter $\hat{\mathcal{R}}_b$ confidence intervals despite the rather larger uncertainties on the $\epsilon_Q^q$ tagging efficiency. As the result, the extracted values of $\hat{\mathcal{R}}_b$ for the two benchmarks are better statistically separated compared to when only the $\{n_b\}$ binned pseudo-data is used. Table II shows that even using suboptimal taggers, with quite likely inflated systematic uncertainties, and without incorporating the full $p_T$ dependence of the $\epsilon_\beta^\alpha$ efficiencies, the model is flexible enough to capture the true distributions of the measured observables $\{n_b, n_q\}$. Furthermore, including $n_q$ as the observable improves the fit sensitivity to $\mathcal{R}_b$. We take this as a starting point to suggest

| Observables | $\mathcal{R}_b = \mathcal{R}_b^{\mathrm{SM}}$ | $\mathcal{R}_b = 0.9$ |
|---|---|---|
| $\{n_b\}$ | [0.894, 1.067] | [0.808, 0.970] |
| $\{n_b, n_q\}$ | [0.978, 1.067] | [0.858, 0.980] |

**Table II:** Estimated 95% confidence intervals for the extracted $\mathcal{R}_b$, obtained using negative log likelihood in Eq. (10), for two input values, $\mathcal{R}_b = \mathcal{R}_b^{\mathrm{SM}} \approx 0.998$ (2nd column), and $\mathcal{R}_b = 0.9$ (3rd column), using either pseudo-data with just $\{n_b\}$ bins (2nd row), or when binned in $\{n_b, n_q\}$ bins (3rd row).

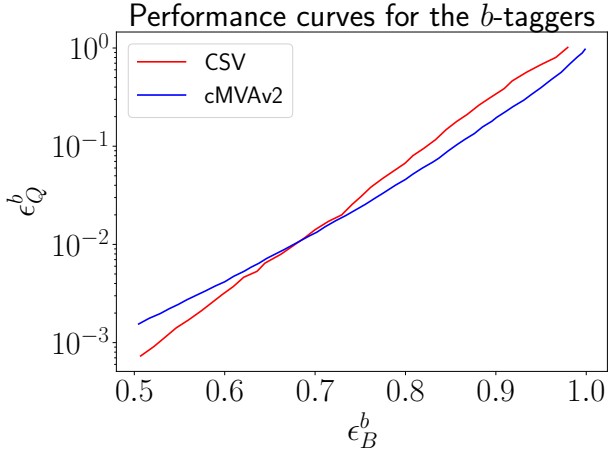

**Figure 4:** Performance curves for the two state of the art $b$-taggers, CSV [12] (red solid line) and cMVAv2 [23] (blue), implemented in the projected sensitivity study.

an improved analysis strategy, which we work out in the next section.

## IV. PROJECTED SENSITIVITY

The improved strategy to measure $\mathcal{R}_b$ includes information on $n_q$ in an optimized way, by using two working points for the $b$-tagger. One of the two working points is used to define an anti-$b$ tagger, which, combined with a quark/gluon-tagger, then defines and improved version of a $q$-tagger. Here, we take advantage of the fact that the state of the art $b$-taggers allow for a greater spectrum of working points, each of which then defines $\{n_b, n_q\}$ bins of varying sample purity. Naively one may expect that the purer the samples the more precise the resulting measurement of $\mathcal{R}_b$ is, given that this leads to the smallest cross-contamination between $n_b$ and $n_q$ variables. However, requiring high false negative rates results in the lack of statistics in the bins with medium to high values of $n_b$ and $n_q$, and consequently to a loss of precision in the extracted value of $\mathcal{R}_b$. This is specially important for realistic values of $\mathcal{R}_b$ close to the SM value, $\mathcal{R}_b \simeq 1$, since these result in very limited statistics in the bins with nonzero value of $n_q$.

In the numerical analysis we consider two state of the art $b$-taggers: for $\sqrt{s} = 8\,\mathrm{TeV}$ collision energy events we use the CSV tagger [12], and for $\sqrt{s} = 13\,\mathrm{TeV}$ the cMVAv2 tagger [23]. The $(\epsilon_B^b, \epsilon_Q^b)$ performance curves for the two $b$-taggers are shown in Fig. 4. To obtain $\{n_b, n_q\}$ bins of varying purity we select two working points, WP1 and WP2, which define two $b$-taggers, $b_1$ and $b_2$, such that

$$\begin{aligned} \epsilon_B^{b_1} &\leq \epsilon_B^{b_2}, \\ \epsilon_Q^{b_1} &\leq \epsilon_Q^{b_2}. \end{aligned} \tag{15}$$

In the numerical analysis $\epsilon_B^{b_{1,2}}$ are varied in the ranges $\epsilon_B^{b_1} \in [0.53, 0.91]$, $\epsilon_B^{b_2} \in [0.59, 0.97]$ for CVS tagger, cf. first row in Fig. 6, and $\epsilon_B^{b_1} \in [0.55, 0.95]$, $\epsilon_B^{b_2} \in [0.59, 0.99]$ for cMVAv2 tagger, cf. second and third rows

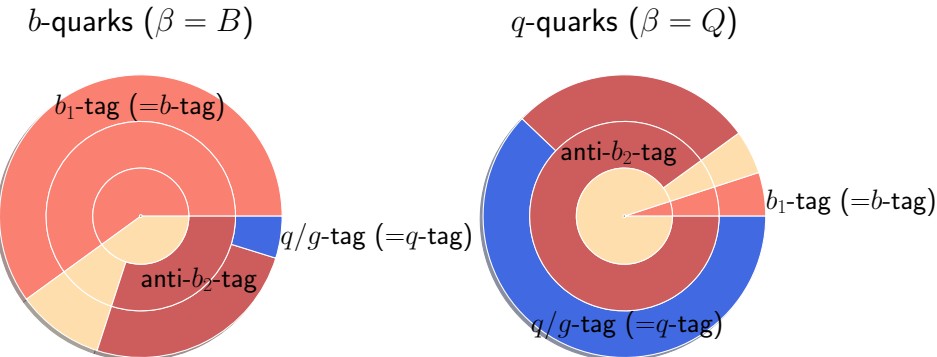

**Figure 5:** Illustrative pie-charts for the tagging procedure of the pseudo-experiments. The left (right) pie-chart reflects how the taggers would apply to truth level $b$-quark ($q$-quark) initiated jets. The specific fractions are not representative.

in Fig. 6. The corresponding efficiencies for the truth level light quark, $\epsilon_Q^{b_{1,2}}$, were extracted from [12, 23] and are shown in Fig. 4. For the $\epsilon_B^{b_1,b_2}$ ranges shown in Fig. 6 , the $\epsilon_Q^{b_1,b_2}$ take values $\epsilon_Q^{b_1} \in [1.1 \cdot 10^{-3}, 0.39]$, $\epsilon_Q^{b_2} \in [2.5 \cdot 10^{-3}, 0.80]$ for the CVS tagger, and $\epsilon_Q^{b_1} \in [2.5 \cdot 10^{-3}, 0.39]$, $\epsilon_Q^{b_2} \in [3.8 \cdot 10^{-3}, 0.81]$ for the cMVAv2 tagger.

The two WP of the taggers are used to sort events into $\{n_b, n_q\}$ bins. First, we apply the $b_1$-tagger, giving $n_b$ $b$-tagged jets for each event. The definition of $n_q$ bins is more involved. In the end we want to obtain a high-purity $q$-tagged jet sample (with almost no $b$ quarks), starting with the jets that were rejected by the $b_1$-tagger. This cannot be achieved by simply applying a quark/gluon-tagger to these jets, since even the state-of-the-art quark/gluon-taggers usually still group together the $b$-quarks and $q$-quarks. However, we can combine the quark/gluon-tagger with an anti-$b$-tagger (using WP2), which gives a $q$-tagger that is orthogonal to the $b$-tagger (from WP1). That is, the $q$-tagged jets belong to the intersection of anti-$b_2$-tagged and quark/gluon-tagged jets, so that the efficiency of the $q$-tagger is $\epsilon_\beta^q = \epsilon_\beta^{\{\text{anti}-b_2\} \cap \{q/g\}}$. For the anti-$b$-tagger we use the WP2 of the $b$-tagger, since the anti-$b_2$ tagger is more powerful in rejecting $b$ quark jets than the anti-$b_1$-tagger is. The $b$- and $q$-taggers defined in this way select non-overlapping $q$- and $b$-tagged jet fractions, i.e., they are orthogonal.

In the numerical analysis we assume for convenience the $q$−tagger efficiencies to be given by

$$\epsilon_Q^q = 0.69(1 - \epsilon_Q^{b_2}), \quad \text{and} \quad \epsilon_B^q = 0.16(1 - \epsilon_B^{b_2}), \tag{16}$$

i.e., that the quark/gluon-tagger always selects a fixed subset of the anti-$b_2$-tagged jets and thus $\epsilon_\beta^q = \epsilon_\beta^{q/g}$. This simplifies our analysis, but it does mean that the quark/gluon tagger working point is also modified for each choice of WP2. With this set-up, the partitioning procedure is as follows: we first $b_1$-tag the objects, obtaining $n_b$ $b_1$-tagged jets. All remaining jets are subjected to the anti-$b_2$-tagger. Finally, we apply a quark/gluon-tagger to all the jets that are anti-$b_2$-tagged, obtaining $n_q$ $q$-tagged jets. A qualitative picture of how the samples are partitioned by this procedure is shown in Fig. 5.

The numerical values in (16) are motivated by the efficiencies quoted in Table I, so that the assumed $q$-tagger efficiencies for this analysis will always be lower than the ones for $q$-tagger in Section III, because of the non-zero anti-$b_2$-tagging efficiency. That $\epsilon_\beta^q$ are varied does have a practical advantage, since we can explore different WP regimes. For low $\epsilon_\beta^{b_2}$ the resulting $q$-tagger will lead to high sample size at the expense of sample purity, while for high $\epsilon_\beta^{b_2}$ sample size decreases considerably as sample purity increases. For the experimental analysis the choices of $\epsilon_\beta^q$ could be further optimized, however, we expect the qualitative conclusions about the added statistical power provided by $\{n_q\}$ binning to be robust.

To assess the sensitivity of the proposed analysis to $\mathcal{R}_b$ we make two simplifying assumptions to speed up

| Nuisance Param. | Uncertainty |
|:---:|:---:|
| $\epsilon_B^{b_1}$ | 2% |
| $\epsilon_Q^{b_1}$ | 11% |
| $\epsilon_{B,Q}^{q}$ | 5% |

**Table III:** Systematic uncertainties for the $b$- and $q$-taggers for the proof-of-concept of the proposed analyses.

the numerics. First, we use $\{n_b, n_q\}$ bins of pseudodata, generated using the probabilistic model, while the actual experimental analysis would use the full implementation of the taggers at the `ROOT` level and recover the expected rates per bin for each working point. This approach is similar to the one taken in the preceding section, except that the probabilistic model is modified to take into account the use of two WPs. The explicit expression for it is given in App. A, Eq. (A31), with the discussion in the paragraph following it. This allows for a very simple implementation of the taggers since we only need to incorporate the reported efficiencies into the probabilistic modelling.

Second, we use the Asimov approximation [14] to evaluate the significance $Z_1$ ("the number of sigmas"), with which the $\mathcal{R}_b = 1$ hypothesis is rejected, when the true value is $\mathcal{R}_b^{\text{SM}}$. That is, we assume that $Z_1$ is given by $Z_1 = \sqrt{q_{1,A}}$, where $q_1$ is the statistics in (13), (14). The value $q_{1,A}$ is obtained from Eq. (14) using the Asimov dataset, i.e., a dataset with each bin yield, $N_{\ell\ell'}$, equal to the expected rate $\bar{N}_{\ell\ell'}$ taking $\mathcal{R}_b = \mathcal{R}_b^{\text{SM}}$ and with the nuisance parameters $\theta_i$ set to their central values, to perform the necessary NLL minimizations. We have verified explicitly the validity of the Asimov approximation for several WPs by performing a set of pseudo-experiments and verifying that the distribution of the test statistics $q_1$ approaches its asymptotic limit, a non-central chi-squared distribution, with the median approximated well by $q_{1,A}$.

We present the results for several center of mass energies $\sqrt{s}$ and luminosities $\mathcal{L}$ in Fig. 6. The first row in Fig. 6 gives the sensitivity one could expect from Run 1 and should be compared with the results in Ref. [3]. The second row gives the sensitivity one can expect from the already available Run 2 data. Finally, the third row gives the projected sensitivity at HL-LHC (albeit using 13 TeV collision energy, instead of 14 TeV). We perform the scans using fixed tagging systematic uncertainties, listed in Table III. The nuisance parameters associated with the $q$-tagger should in general be split into the contributions from the anti-$b_2$-tagger and from the quark/gluon-tagger. However, since we make the simplifying assumption that the quark/gluon tagger always determines the $q$-tagger efficiency, it suffices in this case to vary just the uncertainties associated with $\epsilon_\beta^q = \epsilon_\beta^{q/g}$, disregarding the uncertainties associated with the anti-$b_2$-tagger. The numerical values of the uncertainties we deem to be reasonable benchmarks, and are limiting factors for the achievable significance. This opens the door for further increase in the statistical power of the analysis. To reduce the computational cost of the fit we used the same $\epsilon_{j;\not{t}}^{b,q}$ parameters for all the $\{\ell\ell', n_j\}$ categories, while in a general analysis $\epsilon_{j;\not{t}}^{b,q}$ for each $\{\ell\ell', n_j\}$ category would be floated independently. We have verified for a few cases that the change in the extracted significance due to this simplification is minimal. The true value is set to $\epsilon_{j;\not{t}}^{b,q} = 0.85 \, \epsilon_Q^{b,q}$, which is consistent with the results found in Section III when fitting $\epsilon_{j;\not{t}}^{b,q}$ to the simulation pipeline-generated events. In total, we fit six nuisance parameters for $\{n_b, n_q\}$: four systematic uncertainties associated to $\epsilon_{B,Q}^{b_1,q}$ and the two $\epsilon_{j;\not{t}}^{b,q}$. For $\{n_b\}$ only, the nuisance parameters are three: two systematic uncertainties associated to $\epsilon_{B,Q}^{b_1}$ and $\epsilon_{j;\not{t}}^{b}$. For each $\sqrt{s}$ and $\mathcal{L}$, we report the expected significances using either only $\{n_b\}$ or $\{n_b, n_q\}$. We also compare, for a given $\epsilon_B^{b_1}$, the ratio between said significances using the maximum performance of the $\{n_b, n_q\}$ strategy.

From Fig. 6, we observe how the addition of $n_q$ noticeably increases the significance of the analysis for a wide range of WPs for all three benchmarks. The near-horizontality of the significance evolution shows that $n_q$ is carrying most of the statistical power. We have verified this by computing the significance considering only $\{n_q\}$. However, the complementarity between $n_b$ and $n_q$ is still noticeable and important. For $\sqrt{s} = 8\,\text{TeV}$ there is a relative increase of around 4.5 for most $\epsilon_B^{b_1}$ although the resulting significance is still very low, reflecting the fact that $\mathcal{R}_b^{\text{SM}}$ is indistinguishable from 1 as seen in Section III and Ref. [3]. The power of the analysis increases for medium $\epsilon_B^{b_2}$. When $\epsilon_B^{b_2}$ is high enough, the loss of statistics is too much for the sample purity to compensate.

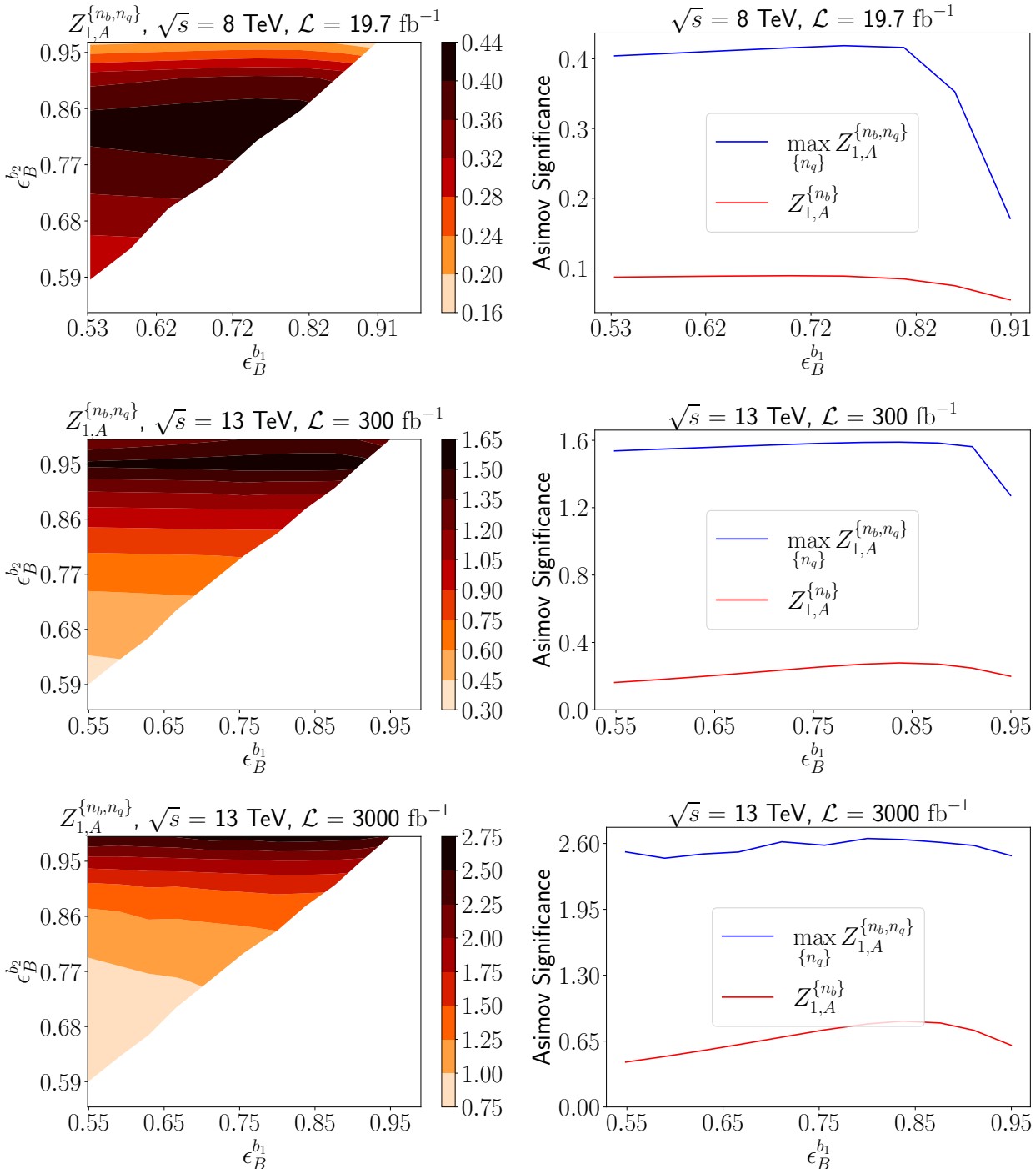

**Figure 6:** Expected results for the proposed strategy using different WPs. Left column: expected discovery Asimov significance for the analysis that uses $\{n_b, n_q\}$ bins, as a function of two $b$-tagging working point efficiencies, $\epsilon_B^{b_1}$, $\epsilon_B^{b_2}$. Right column: Expected discovery Asimov significance for analysis that uses just $\{n_b\}$ bins (red) compared with the highest achievable significance using $\{n_b, n_q\}$ binning. Each row represents a different choice of center-of-mass energy and luminosity, from top to bottom: the 8TeV LHC, the present LHC data, the HL-LHC.

This is no longer the case for the two other benchmarks where the statistics is higher and thus higher $\epsilon_B^{b_2}$ corresponds to higher significance. For the integrated luminosity, $\mathcal{L} = 300\,\mathrm{fb}^{-1}$, the addition of $n_q$ is the difference between being able to exclude the $\mathcal{R}_b = 1$ at above the 1-$\sigma$ level or not. This is already a powerful gain for such a simple modification to the existing strategy. For the HL-LHC with $\mathcal{L} = 3000\,\mathrm{fb}^{-1}$ the expected significance reaches maximum values of around $2.5\sigma$, which is considerably higher than the $0.85\sigma$ achievable with only $\{n_b\}$, and reflects the clear possibility of measuring $\mathcal{R}_b^{\mathrm{SM}}$ directly at the HL-LHC by looking at dileptonic $t\bar{t}$ production. This is achieved by obtaining the highest purity available in the $n_q$ bins, which forces the $\mathcal{R}_b = 1$ hypothesis to push $\epsilon_B^q$ to higher values through its nuisance parameter resulting in a tension with the Asimov dataset.

## V. CONCLUSIONS

We have shown how the probabilistic model implemented in Ref. [3] to measure $\mathcal{R}_b$, Eq. (2), can be extended to incorporate the number of $q$-tagged jets. This additional observable was shown to increase significantly the achievable precision. The proposed measurement strategy incorporates state of the art jet flavor taggers in a way that ensures sample purity, by using two working points. Consequently, the statistical power of the leptonic $t\bar{t}$ LHC dataset is significantly increased, such that one is expected to be able to exclude at HL-LHC the $\mathcal{R}_b = 1$ hypothesis at 95% C.L., and thus show that $|V_{ts}|^2 + |V_{td}|^2 \neq 0$.

There are several ways the present study could be extended. For example, for the proposed strategy in Section IV the pseudo-experiments were performed for each $b$-tagger working point by resorting to the generative model. A more complete implementation would implement the taggers in the full simulation pipeline as done with the simplistic taggers in Section III. A larger set of less crucial systematic uncertainties could also be incorporated. An additional limitation of the presented approach is the absence of jet kinematic information. This could be incorporated in the generative model, similar to the combinatorial likelihood method used, for example, in Ref. [24]. This would come at the cost of a more computationally intensive fit, but with a potentially improved sensitivity to $\mathcal{R}_b$. In short, the presented study makes a compelling case that the addition of $q-$jet information in the experimental analysis of semileptonic $t\bar{t}$ events can greatly increase the precision of $\mathcal{R}_b$ measurement, bringing its SM value within reach of the LHC.

## ◆ ◆ ◆ ACKNOWLEDGMENTS ◆ ◆ ◆

DAF received funding from the European Research Council (ERC) under the European Union's Horizon 2020 research and innovation programme under grant agreement 833280 (FLAY), and by the Swiss National Science Foundation (SNF) under contract 200020-204428. JFK and MS acknowledge the financial support from the Slovenian Research Agency (grant No. J1-3013 and research core funding No. P1-0035). JZ acknowledges support in part by the DOE grant de-sc0011784 and NSF OAC-2103889. The authors are grateful to the Mainz Institute for Theoretical Physics (MITP) of the Cluster of Excellence PRISMA$^+$ (Project ID 39083149), for its hospitality and support.

## Appendix A: Further details on the probabilistic model

In this Appendix we provide explicit expressions for the probability $P_{\ell\ell'}(n_b, n_q | n_j, \mathcal{R}_b, \theta_i)$ in Eq. (3). In the derivation we focus on a particular $\{\ell\ell', n_j\}$ category, with $N$ measured events that are split into $\{n_b, n_q\}$ bins. Here, $n_b = 0, \ldots, n_j$ denotes the number of $b$-quark tagged jets and $n_q = 0, \ldots, n_j - n_b$ the number of light quark tagged jets. The final result is given in Eq. (A31), and is obtained by introducing a number of intermediate sub-categories in order to make a more fine-grained classification of the events.

The flow chart showing the interdependency of the intermediate sub-categories is shown in Fig. 1. The goal is to be able to predict the expected number of events in each of the $\{n_b, n_q\}$ bins. To achieve it one needs to also predict the expected number of events in each of the sub-categories. These subcategories are constrained given the specified $\{n_j, n_b, n_q\}$, and are

- The number of jets $n_j$ is divided in jets originating from top-quark-decays, $n_{j;t}$, and in jets originating from ISR/FSR + background processes, $n_{j;\not{t}} = n_j - n_{j;t}$.

- The $n_{j;t}$ bin is further divided according to the decay channels of the top, using the $n_B$ and $n_Q$ variables. Here, $n_B$ is the number of $b-$quark initiated jets, while $n_Q = n_{j;t} - n_B$ is the number of $d-$ and $s-$quarks initiated jets. When we refer to true $b-$ or $q-$quark jets, we refer to the quark jets originating from top-quark-decays. All jets originating from ISR/FSR + background process are referred to as "$j; \not{t}$", regardless of the initial parton.

- Each of the three truth level jet types, $\beta = \{B, Q, j; \not{t}\}$, can be either $b$- or $q$-tagged, populating the $n_{b,q;\beta}$ subcategories. Because the taggers are orthogonal, a single jet can populate at most one subcategory. If the jet is neither $b-$ nor $q-$tagged, it populates a separate subcategory, not shown in the graphical model, since it provides no additional information. For a given $\beta$, we have then $n_{b;\beta}$ and $n_{q;\beta}$ with $n_{b,\beta} + n_{q;\beta} \leq n_\beta$. For simplicity, we denote the number of $\alpha-$tagged "$j; \not{t}$" jets as $n_{\alpha;\not{t}}$.

- We group together $n_{\alpha;Q}$ and $n_{\alpha;B}$ into the number of $\alpha-$tagged jets originating from top-quark-decays $n_{\alpha;t}$. The $n_\alpha$, $n_{\alpha;\not{t}}$ and $n_{\alpha;t}$ variables satisfy $n_{\alpha;\not{t}} = n_\alpha - n_{\alpha;t}$.

The sub-categories in black circles in Fig. 1 were already used in the inference model of Ref. [25], while the red encircled sub-categories are new. The arrows in Fig. 1 denote the probabilities for splitting the events into particular sub-categories. For instance, for a single event with $n_j$ jets, the probability, $P(n_{j;t}|n_j)$, that $n_{j;t}$ out of $n_j$ observed jets originate from top-quark-decays, is given by[2]

$$P(n_{j;t}|n_j) = \sum_{z=t\bar{t},\mathrm{st},\mathrm{bkg}} P(n_{j;t}|z)P(z|n_j), \tag{A1}$$

where the summation is over all three event types: $pp \to t\bar{t}$, single top and background events. Here $P(z|n_j)$ denotes the probability for an event in the $\{\ell\ell', n_j\}$ category to belong to the event type $z = \{t\bar{t}, \mathrm{st}, \mathrm{bkg}\}$, while $P(n_{j;t}|z)$ gives a probability for a given event type to have $n_{j;t}$ observed top decay jets. Below we derive expressions for both types of probabilities. In Eq. (A2) the $P(z|n_j)$ are expressed in terms of two nuisance parameters, while $P(n_{j;t}|z)$ are given in (A4).

We focus first on $P(z|n_j)$ and write

$$P(tt|n_j) = f_{t\bar{t}}, \qquad P(\mathrm{st}|n_j) = k_{\mathrm{st}} f_{t\bar{t}}, \qquad P(\mathrm{bkg}|n_j) = 1 - P(t\bar{t}|n_j) - P(\mathrm{st}|n_j). \tag{A2}$$

The nuisance parameters $f_{t\bar{t}}$ and $k_{\mathrm{st}}$ are determined in the following way. In each $\{\ell\ell', n_j\}$ category first the signal strength $\hat{\mu}$ is determined by summing over all the $\{n_b, n_q\}$ bins, and comparing the observed total number of events, $N_{\ell\ell'}(n_j)$, with the expected number of events, $\bar{N}_{\ell\ell'}^{\mathrm{MC}}(n_j)$, that was obtained using Monte Carlo, see Section. III for details on the Monte Carlo pipeline[3]. Both in $N_{\ell\ell'}(n_j)$ and the expected number of events we sum over all three event types, $z = \{t\bar{t}, \mathrm{st}, \mathrm{bkg}\}$. The signal strength $\hat{\mu} = N_{\ell\ell'}(n_j)/\bar{N}_{\ell\ell'}^{\mathrm{MC}}(n_j)$ is then traded for the nuisance parameter $f_{t\bar{t}}$, denoting the fraction of $t\bar{t}$ events. We determine its value through the relation $f_{t\bar{t}} = \hat{\mu}\bar{N}_{t\bar{t}}^{\mathrm{MC}}/N$, where $\bar{N}_{t\bar{t}}^{\mathrm{MC}}$ is the expected number of $t\bar{t}$ events obtained using Monte Carlo (for shortness we are dropping the $\ell\ell'$ and $n_j$ labels for the remainder of this section).

For $\mathcal{R}_b = 1$ the relative fraction of single top events, $k_{\mathrm{st}}$, is determined through $k_{\mathrm{st}}(\mathcal{R}_b = 1) = \bar{N}_{\mathrm{st}}^{\mathrm{MC}}/\hat{\mu}\bar{N}_{t\bar{t}}^{\mathrm{MC}}$, where $\bar{N}_{\mathrm{st}}^{\mathrm{MC}}$ is the number of single top events expected from Monte Carlo assuming $\mathcal{R}_b = 1$. Single top production fraction for arbitrary $\mathcal{R}_b$ is then, assuming CKM unitarity and $tW$ dominance in production of a single top, given by [25],

$$\frac{k_{\mathrm{st}}(\mathcal{R}_b)}{k_{\mathrm{st}}(\mathcal{R}_b = 1)} \approx \mathcal{R}_b + \frac{1 - \mathcal{R}_b}{1 + |V_{ts}/V_{td}|^2}\left(\frac{|V_{ts}|^2}{|V_{td}|^2}\frac{\sigma_d^{tW}}{\sigma_b^{tW}} + \frac{\sigma_s^{tW}}{\sigma_b^{tW}}\right), \tag{A3}$$

where the Monte Carlo computed single top cross-sections with initial $d$-, $s$- and $b$-quarks are denoted as $\sigma_{d,s,b}^{tW}$, respectively. We are also neglecting any difference in efficiencies and acceptances between different $tW$ production processes.

---

[2] Here and below all the probabilities are assumed to in general depend on the flavor $\ell\ell'$ of the final state, while we do not display this dependence explicitly for brevity.

[3] We denote the measured values for events in each of subcategories with $N$, the expected values using the probabilistic model with $\bar{N}$, and the expected values that use just Monte Carlo, with $\bar{N}^{\mathrm{MC}}$.

We turn next to $P(n_{j;t}|z)$, the probability for a given event type to have $n_{j;t}$ observed jets that originate from top decays. It is given by

$$P(n_{j;t}|z) = \text{Binom}(n_{j;t}, n_{t;z}, p) \equiv \binom{n_{t;z}}{n_{j;t}} p^{n_{j;t}} (1-p)^{n_{t;z}-n_{j;t}}, \tag{A4}$$

with $p$ given in Eq. (A7) below. Here, $n_{t;z} = \{2, 1, 0\}$ is the number of tops in a $z = \{t\bar{t}, \text{st}, \text{bkg}\}$ event type, respectively. The binomial symbol in (A4) is understood to vanish for $n_{t;z} < n_{j;t}$. For instance, for the background events we thus have $P(n_{j;t}|\text{bkg}) = 0$ for $n_{j;t} = 1, 2$, and $P(n_{j;t} = 0|\text{bkg}) = 1$.

The probability $p$ in Eq. (A7) denotes the probability of capturing the top decay products. Experimentally, the tops are identified on a statistical basis by forming lepton–jet pairs from the top decay products, and therefore we define $p$ as (for each $\{\ell\ell', n_j\}$ category)

$$p = \frac{N_{\ell j;t}}{\bar{N}_{\ell j;t}}, \tag{A5}$$

where $N_{\ell j;t}$ ($\bar{N}_{\ell j;t}$) is the measured (expected) number of lepton-jet pairs where the jet comes from the same decaying top as the lepton and we sum over both $pp \to t\bar{t}$ and single top processes. Since it is impossible to identify from data on an event by event basis that the jet definitely originated from a top, the $N_{\ell j;t}$ is not directly observable, and is "measured" only in the sense that it can be determined from data with some further modeling input. Denoting by $N_{\ell j}$ the number of all lepton–jet pairs that can be constructed from the measured events, we first introduce

$$f_{\ell j;t} = \frac{N_{\ell j;t}}{N_{\ell j}}, \tag{A6}$$

which denotes a fraction of all possible lepton–jet pairs that are due to top decays, and furthermore have a lepton and a jet both correctly assigned to the mother top (if the jet indeed originated from a top decay). For instance, for just one $pp \to t\bar{t}$ event, with both tops decaying semileptonically and both $b$−jets observed by the experiment, we would have $N_{\ell j;t} = 2$, $N_{\ell j} = 4$ and thus $f_{lj;t} = 0.5$. In general, the value of $f_{lj;t}$ depends on the kinematical cuts, through the changed fractions of $z = \{t\bar{t}, \text{st}, \text{bkg}\}$ event type fractions, as well as on experimental efficiencies. For instance, in our Monte Carlo study the value of $f_{lj;t}$ would be given by the ratio between the number of events denoted with blue line in Fig. 7, and the number of events contained in the gray histogram.

Importantly, $f_{lj;t}$ can be determined via data driven methods with minimal modeling assumptions. We follow the procedure introduced in Refs. [3, 26, 27], which is based on $t\bar{t}$ kinematics. The spectrum of lepton–jet pairs as a function of their invariant mass, $m_{\ell j}$, is shown in Fig. 7 with a blue line. At the parton level the $m_{\ell j}$ distribution would have an end-point at $m_{\ell j} \leq \sqrt{m_t^2 - m_W^2} \approx 153 \, \text{GeV}$, which due to jet algorithms effects gets somewhat smeared in the experiment. The bulk of the $m_{\ell j}$ distribution lies below the $\sqrt{m_t^2 - m_W^2}$ value, denoted with a dashed vertical line in Figs. 7 and 8, while the $m_{\ell j}$ spectrum of all the possible lepton-jet pairs (grey histrogram) has a long tail well above the dashed line.

We can use this feature to construct a *jet misassignment model* and determine $f_{\ell j;t}$ from data. We assume that the events in the $dN_{\ell j}/dm_{\ell j}$ distribution with $m_{\ell j} > 180$ GeV contain no correctly assigned lepton-jet pairs that would originate from top decays. The mass cut is shown with dotted vertical line in Figs. 7 and 8, which shows that this is a very good approximation. The remaining misassigned events are thus either due to a jet that originated from the other top, or due to jets from backgrounds. We then model the misassignment of lepton–jet pairs by taking all the data and rotate randomly the $(\cos\theta, \phi)$ variables of each lepton, constructing all possible lepton–jet pairs. This gives the red histogram in Fig. 7. The validity of this approximation relies on the fact that the lepton and jet are statistically uncorrelated as long as they do not originate from the same top-quark-decay.

The ratio of the events in grey and red histograms above the mass cut is our estimate of $1 - f_{\ell j;t}$. This data driven method approximates well the true $f_{\ell j;t}$, as can be seen from Fig. 8. The data in the red histogram in Fig. 7, rescaled by $1 - f_{\ell j;t}$, gives the red dots. These agree well with the simulated data shown with black dots (equivalent to grey histogram in Fig. 7) above the mass cut. Subtracting the estimated spectrum of misassigned lepton-jet pairs from $dN_{\ell j}/dm_{\ell j}$ then gives the model for the $dN_{\ell j;t}/dm_{\ell j}$, shown with magenta in Fig. 8, which agrees well with the truth level distribution (blue points). Finally, we can write for the $p$

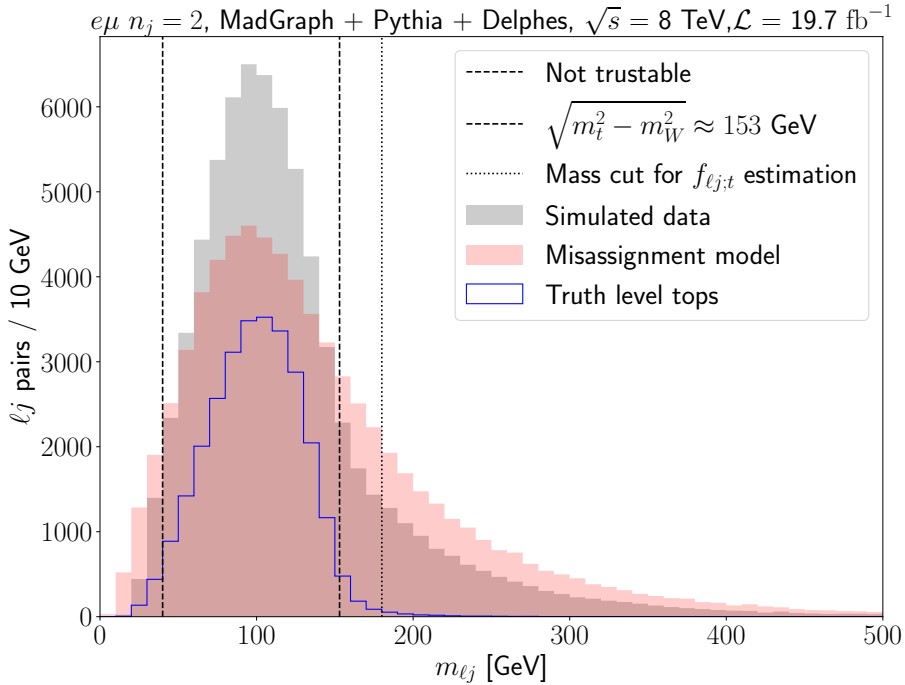

**Figure 7:** Inclusive lepton-jet invariant mass $m_{\ell j}^{\text{incl.}}$ spectrum (black), post-fit jet misassignment model (red) and correctly assigned lepton-jet pair at truth level (blue). See text for further details.

probability,

$$p = \frac{N_{\ell j;t}}{\bar{N}_{\ell j;t}} = \frac{f_{\ell j;t} N_{\ell j}}{\bar{N}_{\ell j;t}} = \frac{f_{\ell j;t} 2 n_j N_{\ell\ell'}(n_j)}{(2 f_{t\bar t} + f_{t\bar t} k_{\text{st}}) N_{\ell\ell'}(n_j)} = \frac{f_{\ell j;t} n_j}{(f_{t\bar t} + f_{t\bar t} k_{\text{st}}/2)}, \tag{A7}$$

where $k_{\text{st}}$ depends on $\mathcal{R}_b$ through (A3), while $f_{\ell j;t}$ and $f_{t\bar t}$ do not.

Explicitly, the probabilities $P(n_{j;t}|n_j)$, Eq. (A1), are given by

$$P(n_{j;t} = 0|n_j) = (1-p)^2 f_{t\bar t} + (1-p) f_{t\bar t} k_{\text{st}}(\mathcal{R}_b) + \left[1 - f_{t\bar t}(1 + k_{\text{st}}(\mathcal{R}_b))\right], \tag{A8}$$

$$P(n_{j;t} = 1|n_j) = 2p(1-p) f_{t\bar t} + p f_{t\bar t} k_{\text{st}}(\mathcal{R}_b), \tag{A9}$$

$$P(n_{j;t} = 2|n_j) = p^2 f_{t\bar t}. \tag{A10}$$

One can verify that $\sum_{n_{j;t}} P(n_{j;t}) = 1$.

The next step in the flow chart of the probabilistic model in Fig. 1 is to consider all possible origins of the $b$-tagged jets. For an event in the $n_{j;t}$ subcategory the probability to have $n_b$ $b$-tagged jets is given by

$$P(n_b|n_{j;t}) = \sum_{n_{b;t}=0}^{\min(n_b, n_{j;t})} P(n_{b;t}|n_{j;t}) P(n_{b;\slashed{t}}|n_{j;\slashed{t}}), \tag{A11}$$

where $P(n_{b;t}|n_{j;t})$ is the probability that out of $n_{j;t}$ top quark originated jets, $n_{b;t}$ are tagged as $b$-jets. The factor $P(n_{b;\slashed{t}}|n_{j;\slashed{t}})$ gives the probability for the remaining $b$-tagged jets not to originate from top quark decays. Here, $n_{b;\slashed{t}} = n_b - n_{b;t}$ is the number of $b$−tagged jets not coming from a top decay, while $n_{j;\slashed{t}} = n_j - n_{j;t}$ is the number of jets not coming from top decays. In terms of mistagging efficiencies $\epsilon_{j;\slashed{t}}^b$ we have

$$P(n_{b;\slashed{t}}|n_{j;\slashed{t}}) = \text{Binom}(n_{b;\slashed{t}}, n_{j;\slashed{t}}, \epsilon_{j;\slashed{t}}^b), \tag{A12}$$

where $\epsilon_{j;\slashed{t}}^b$ depends on the category $\{\ell\ell', n_j\}$, which we suppress in the notation, as always.

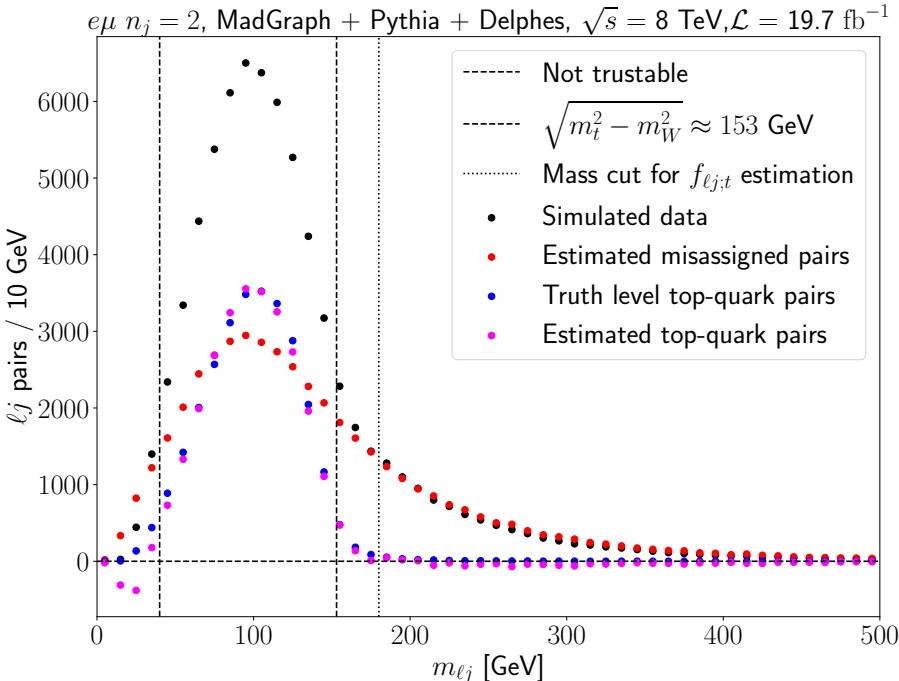

**Figure 8:** Inclusive lepton-jet invariant mass $m_{\ell j}^{\text{incl.}}$ spectrum (black), post-fit jet misassignment model (red), correctly assigned lepton-jet pair at truth level (blue) and extracted lepton-jet pairs from top-quark decays (magenta). See text for further details.

The explicit dependence on $\mathcal{R}_b$ enters through $P(n_{b;t}|n_{j;t})$. We need to distinguish between $b$-quarks and $q$-quarks coming from the top, so that we write

$$P(n_{b;t}|n_{j;t}) = \sum_{n_B=0}^{n_{j;t}} P(n_{b;t}|n_B, n_{j;t})P(n_B|n_{j;t}), \tag{A13}$$

where the summation is over $n_B$, the number of true $b-$quark jets coming from top decays. The probability for having $n_B$ true $b-$quarks in the event with $n_{j;t}$ jets originating from top decays, is given by

$$P(n_B|n_{j;t}) = \text{Binom}(n_B, n_{j;t}, \mathcal{R}_b). \tag{A14}$$

The probability of obtaining $n_{b;t}$ $b$-tagged jets given $n_B$ true $b-$quarks coming from top decays, on the other hand, is given by

$$P(n_{b;t}|n_B, n_{j;t}) = \sum_{n_{b;B}=0}^{\min(n_{b;t}, n_B)} P(n_{b;B}|n_B)P(n_{b;Q}|n_Q), \tag{A15}$$

where $P(n_{b;B}|n_B)$ gives the probability of having $n_{b;B}$ $b-$tagged jets, originating from $n_B$ true hard $b$ quarks, while $P(n_{b;Q}|n_Q)$ gives the probability for $n_{b:Q} = n_{b;t} - n_{b;B}$ $b$-tagged jets to come from $n_Q = n_{j;t} - n_B$ true $q-$quark jets coming from top decays. Explicitly, these are given by

$$P(n_{b;B}|n_B) = \text{Binom}(n_{b;B}, n_B, \epsilon_B^b), \tag{A16}$$

$$P(n_{b;Q}|n_Q) = \text{Binom}(n_{b;Q}, n_Q, \epsilon_Q^b), \tag{A17}$$

and the summation in (A14) is over $n_{b;B}$, the number of $b$-tagged jets originating from a true $b$ quark.

Collecting all the results so far, we can write down the explicit expression for probability $P(n_b|n_j)$, i.e., the probability for $n_b$ $b-$tagged jets in an event with $n_j$ jets,

$$P(n_b|n_j) = \sum_{n_{j;t}=0}^{2} P(n_b|n_{j;t})P(n_{j;t}|n_j). \tag{A18}$$

Resolving $P(n_b|n_{j;t})$ in terms of probabilities involving sub-categories we obtain,

$$\begin{aligned}
P(n_b|n_j) &= \sum_{n_{j;t}=0}^{2} P(n_b|n_{j;t})P(n_{j;t}|n_j) = \sum_{n_{j;t}=0}^{2} P(n_{j;t}|n_j) \sum_{n_{b;t}=0}^{\min(n_b,n_{j;t})} P(n_{b;t}|n_{j;t})P(n_{b;\not t}|n_{j;\not t}) \\
&= \sum_{n_{j;t}=0}^{2} P(n_{j;t}|n_j) \sum_{n_{b;t}=0}^{\min(n_b,n_{j;t})} P(n_{b;\not t}|n_{j;\not t}) \sum_{n_B=0}^{n_{j;t}} P(n_{b;t}|n_B, n_{j;t})P(n_B|n_{j;t}) \\
&= \sum_{n_{j;t}=0}^{2} P(n_{j;t}|n_j) \sum_{n_{b;t}=0}^{\min(n_b,n_{j;t})} P(n_{b;\not t}|n_{j;\not t}) \sum_{n_B=0}^{n_{j;t}} P(n_B|n_{j;t}) \times \\
&\qquad\qquad\qquad\qquad \times \sum_{n_{b;B}=0}^{\min(n_{b;t},n_B)} P(n_{b;B}|n_B)P(n_{b;Q}|n_Q).
\end{aligned} \tag{A19}$$

Using the results in Eqs. (A12), (A14), (A16), (A17), we now obtain the expression for $P(n_b|n_j)$ in terms of the parameters $\mathcal{R}_b, \epsilon_B^b, \epsilon_Q^b, \epsilon_{j;\not t}^b$, as well as the sub-category labels that we sum over,

$$\begin{aligned}
P(n_b|n_j) &= \sum_{n_{j;t}=0}^{2} P(n_{j;t}|n_j) \sum_{n_B=0}^{n_{j,t}} \mathrm{Binom}(n_B, n_{j;t}, \mathcal{R}_b) \times \\
&\qquad \times \sum_{n_{b;t}=0}^{\min(n_b,n_{j;t})} \mathrm{Binom}(n_{b;\not t}, n_{j;\not t}, \epsilon_{j;\not t}^b) \times \\
&\qquad \times \sum_{n_{b;B}=0}^{\min(n_{b;t},n_B)} \mathrm{Binom}(n_{b;B}, n_B, \epsilon_B^b)\, \mathrm{Binom}(n_{b;Q}, n_Q, \epsilon_Q^b),
\end{aligned} \tag{A20}$$

where $P(n_{j;t}|n_j)$ are given in Eqs. (A8)-(A10).

The probabilistic model in Eq. (A20) was used in Ref. [25] to place bounds on $\mathcal{R}_b$. We now modify it to include the additional sub-categories in Fig. 1. We first consider the simpler case of just a single $b$-tagger working point and a single $q$-tagger working point, i.e., the case that was considered in Section III. Following the same approach as for $P(n_b|n_j)$ in (A19), we can write out the probability $P(n_b, n_q|n_j)$, in terms of the nested probabilities for the sub-categories. Here $P(n_b, n_q|n_j)$ is the probability for an event with $n_j$ jets to have out of these $n_b$ jets $b-$ tagged and $n_q$ jets to be $q$-tagged. Instead of (A18) we now have

$$P(n_b, n_q|n_j) = \sum_{n_{j;t}=0}^{2} P(n_b, n_q|n_{j;t})P(n_{j;t}|n_j). \tag{A21}$$

The $P(n_{j;t}|n_j)$ are still given in terms of Eqs. (A8)-(A10), while for $P(n_b, n_q|n_{j;t})$, i.e, the probability to have $n_b(n_q)$ $b(q)$-tagged jets given $n_{j;t}$ jets originating from top decays, we can write

$$\begin{aligned}
P(n_b, n_q|n_{j;t}) = \sum_{n_B=0}^{n_{j;t}} \sum_{n_{b;t}=0}^{\min(n_{j;t},n_b)} \sum_{n_{q;t}=0}^{\min(n_{j;t}-n_{b;t},n_q)} \\
\sum_{n_{b;B}=0}^{\min(n_{b;t},n_B)} \sum_{n_{q;B}=0}^{\min(n_{q;t},n_B-n_{b;B})} P(n_b, n_q, n_B, n_{b;t}, n_{q;t}, n_{b;B}, n_{q;B}|n_{j;t}),
\end{aligned} \tag{A22}$$

where the summation is over the sub-category labels. Out of these, two are new: $n_{q;t}$ denotes the number of $q$-tagged jets that originated from decays of top-quarks, and $n_{q;B}$, which denotes the number of true $b$-quarks in the event that are mistagged as $q$ jets (i.e., are $q-$tagged). The other sub-category labels are the same as before: $n_B$ is the number of true $b$-quarks originating from a top-quark decay, $n_{b;t}$ is the number of $b$-tagged top-quark originated jets, and $n_{b;B}$ the number of $b$-tagged true $b$-quarks in the event. The probability for an event with $n_{j,t}$ jets coming from top decays to be in the $\{n_b, n_q, n_B, n_{b;t}, n_{q;t}, n_{b;B}, n_{q;B}\}$ sub-category can be further decomposed as

$$P(n_b, n_q, n_B, n_{b;t}, n_{q;t}, n_{b;B}, n_{q;B}|n_{j;t}) = P(n_b, n_q, n_{b;t}, n_{q;t}, n_{b;B}, n_{q;B}|n_{j;t}, n_B)P(n_B|n_{j;t}), \tag{A23}$$

with $P(n_B|n_{j;t})$ given in (A14). We can further distinguish between tagged jets originating from top-quark-decays and from ISR/FSR + background, and write

$$P(n_b, n_q, n_{b;t}, n_{q;t}, n_{b;B}, n_{q;B}|n_{j;t}, n_B) = P(n_{b;\not t}, n_{q;\not t}|n_{j;\not t})P(n_{b;t}, n_{q;t}, n_{b;B}, n_{q;B}|n_{j;t}, n_B), \tag{A24}$$

where the first term is the probability of $n_{b;\not t} = n_b - n_{b;t}$ $b$-tagged jets and $n_{q;\not t} = n_q - n_{q;t}$ $q$-tagged jets originating from $n_{j;\not t} = n_j - n_{j;t}$ jets that do not originate from top-quark-decays and the second term is the probability of obtaining $n_{b;t}$ $b$-tagged jets and $n_{q;t}$ $q$-tagged jets from $n_{j;t}$ jets originating from top-quark-decays. We can expand the latter probability further, by requiring to distinguish between true $q$–quarks and true $b$–quarks

$$P(n_{b;t}, n_{q;t}, n_{b;B}, n_{q;B}|n_{j;t}, n_B) = P(n_{b;Q}, n_{q;Q}|n_Q)P(n_{b;B}, n_{q;B}|n_B). \tag{A25}$$

We have three different sources of $b$-tagged and $q$-tagged jets: true $b$–quarks, $n_B$, true $q$–quarks, $n_Q$ and jets that do not originate from top-quark-decays, $n_{j;\not t}$. For all three possibilities we introduce the joint probabilities $P(n_{b;\beta}, n_{q;\beta}|n_\beta)$, where $\beta = \{B, Q, j; \not t\}$. Although the $b-$tagger and $q-$tagger are orthogonal, the shared pool of $n_\beta$ jets from which one tags introduces a non-trivial structure in the joint distributions. We expand the joint probabilities following the product rule and write

$$P(n_{b;\beta}, n_{q;\beta}|n_\beta) = P(n_{b;\beta}|n_\beta)P(n_{q;\beta}|n_{b;\beta}, n_\beta), \qquad \text{for} \quad \beta = \{B, Q, j; \not t\}. \tag{A26}$$

The first term is the same as in Eq. (A16). The second term is also a binomial distribution, but it is modified by the conditioning on $n_{b;\beta}$. First, the available number of jets to tag is reduced from $n_\beta$ to $n_{\beta;\not b} = n_\beta - n_{b;\beta}$ due to $n_{b;\beta}$ jets already being $b$-tagged. Second, the probability of tagging a single jet is modified because one considers a different ensemble of jets

$$\frac{\bar{N}_{q;\beta}}{\bar{N}_\beta} = \epsilon_\beta^q \to \frac{\bar{N}_{q;\beta}}{\bar{N}_{\beta;\not b}} = \frac{\bar{N}_{q;\beta}}{\bar{N}_\beta - \bar{N}_{b;\beta}} = \frac{\epsilon_\beta^q}{1 - \epsilon_\beta^b}. \tag{A27}$$

Because of Eq. (6), the efficiency ratios $\epsilon_\beta^q/(1 - \epsilon_\beta^b)$ are well behaved. Collecting the intermediate results, we obtain

$$P(n_{b;\beta}, n_{q;\beta}|n_\beta) = \text{Binom}(n_{b;\beta}, n_\beta, \epsilon_\beta^b)\,\text{Binom}(n_{q;\beta}, n_\beta - n_{b;\beta}, \epsilon_\beta^q/(1 - \epsilon_\beta^b)). \tag{A28}$$

Using Eq. (A28) in the first term of Eq. (A24), we obtain

$$\begin{aligned} P(n_{b;\not t}, n_{q;\not t}|n_{j;\not t}) = \text{Binom}(n_{b;\not t}, n_{j;\not t}, \epsilon_{j;\not t}^b) \times \\ \times \text{Binom}(n_{q;\not t}, n_{j;\not t} - n_{b;\not t}, \epsilon_{j;\not t}^q/(1 - \epsilon_{j;\not t}^b)), \end{aligned} \tag{A29}$$

while the second term yields

$$\begin{aligned} P(n_{b;t}, n_{q;t}, n_{b;B}, n_{q;B}|n_{j;t}, n_B) = \text{Binom}(n_{b;B}, n_B, \epsilon_B^b)\,\text{Binom}(n_{q;B}, n_B - n_{b;B}, \epsilon_B^q/(1 - \epsilon_B^b)) \times \\ \times \text{Binom}(n_{b;Q}, n_Q, \epsilon_Q^b)\,\text{Binom}(n_{q;Q}, n_Q - n_{b;Q}, \epsilon_Q^q/(1 - \epsilon_Q^b)). \end{aligned} \tag{A30}$$

Grouping it all together we obtain

$$
\begin{aligned}
P\big(n_b, n_q | n_j\big) = \sum_{n_{j;t}=0}^{2} P_{\ell\ell'}(n_j) \sum_{n_B=0}^{n_{j;t}} \text{Binom}\big(n_B, n_{j;t}, \mathcal{R}_b\big) \sum_{n_{b;t}=0}^{\min(n_b, n_{j;t})} \text{Binom}\big(n_{b;\slashed{t}}, n_{j;\slashed{t}}, \epsilon_{j;\slashed{t}}^{b}\big) \times \\
\times \sum_{n_{q;t}=0}^{\min(n_q, n_{j;t}-n_{b;t})} \text{Binom}\big(n_{q;\slashed{t}}, n_{j;\slashed{t}} - n_{b;\slashed{t}}, \epsilon_{j;\slashed{t}}^{q}/(1-\epsilon_{j;\slashed{t}}^{b})\big) \times \\
\times \sum_{n_{b;B}=0}^{\min(n_{b;t}, n_B)} \text{Binom}(n_{b;B}, n_B, \epsilon_B^b)\, \text{Binom}(n_{b;Q}, n_Q, \epsilon_Q^b) \times \\
\times \sum_{n_{q;B}=0}^{\min(n_{q;t}, n_B-n_{b;B})} \text{Binom}\big(n_{q;B}, n_B - n_{b;B}, \epsilon_B^q/(1-\epsilon_B^b)\big) \times \\
\times \text{Binom}\big(n_{q;Q}, n_Q - n_{b;Q}, \epsilon_Q^q/(1-\epsilon_Q^b)\big),
\end{aligned}
\tag{A31}
$$

to be used in (A21).

In Section IV we developed an improved strategy to probe $\mathcal{R}_b$, by using two working points of a state-of-the-art $b$-tagger, WP1 and WP2, see Eq. (15). The $b_1$ working point is used as a $b$-tagger, while $b_2$ working point is used as an anti-$b$-tagger, which in combination with a quark/gluon tagger defines a $q$-tagger. The use of two working points increases the sample purity and consequently the sensitivity of the analysis to nonzero value of $|V_{td}|^2 + |V_{ts}|^2$. The probabilistic model for the analysis proposed in Section IV is still given by Eq. (A31), but replacing $\epsilon_\beta^b$ with $\epsilon_\beta^{b_1}$ and $\epsilon_\beta^q/(1-\epsilon_\beta^b)$ to $\epsilon_\beta^q/(1-\epsilon_\beta^{b_1})$ (here, the second working point and the quark/gluon tagger are implicit in the $q$-tagger definition). The introduction of an additional working point increases the number of systematic uncertainties for $\epsilon_{B,Q}^\alpha$, where now $\alpha = b_1, b_2, q/g$. In a general analysis, the efficiencies $\epsilon_{B,Q}^{b_2}$ and $\epsilon_{B,Q}^{q/g}$ vary and modify $\epsilon_{B,Q}^q$. However, within our simplifying assumptions this is not the case, because the quark/gluon-tagger is always a subset of the anti-$b_2$-tagger, and thus we only need to vary uncertainties on $\epsilon_{B,Q}^{q/g}(= \epsilon_{B,Q}^q)$, see Section IV. The $\epsilon_{j;\slashed{t}}^\alpha$ efficiencies, on the other hand, continue to be fitted from data. No modification is therefore needed in the part of Eq. (A31) which deals with jets that do not originate from top-quark-decays.

Further modifications of the probabilistic model are possible. One could, for instance, incorporate the $p_T$ dependence of the taggers to increase the ability of the model to capture the true probability distribution. This is achievable by introducing latent variables at the expense of higher computational cost when minimizing the negative log-likelihood. This in turn could be offset by turning to other algorithms such as expectation-minimization or variational inference. We leave such modifications for future work.

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
