# Peer review of "Accessing CKM suppressed top decays at the LHC"

_SciPost Physics_

## Round 1 · Referee Report · Anonymous (Referee 1) · 2022-10-27

Strengths

  1. This is an interesting paper showing that to measure Vtb different from one counting q-jets in addition to counting b-jets substantially improves the sensitivity, potentially making a measurement possible at HL-LHC.
  2. The paper is very well-written. The structure is clear and all the information needed to fully understand the paper is available.
  3. The authors also clearly identify potential weaknesses in their conclusions. All the potential issues seem to have been identified.

Weaknesses

  1. Related to the identified weaknesses, some statements should probably be rephrased as it doesn't seem certain the result of a more realistic model will always go in the direction assumed by the authors.

Report

This is an interesting paper showing that to measure Vtb different from one counting q-jets in addition to counting b-jets substantially improves the sensitivity, potentially making a measurement possible at HL-LHC. The conclusions of the paper seem sound even if the model is simplified. This paper is definitely worthy of publication.

Requested changes

  1. The taggers don't need to be orthogonal for a jet to get a single tag. Eg one could cascade two taggers (as is done in the paper): b or not, then apply a q/g discriminator to the not-b events. These two taggers need not be orthogonal, but the desired output is achieved. (The case presented in section 4 in fact uses non-orthogonal taggers.)

  2. It is not usually true that the tagging efficiencies for b jets are independent of the number of jets in the event. But this is a correction factor that can be applied by experiments.

  3. Towards the bottom of p3 the text now seems to have alpha as lower index and beta as higher. This is confusing.

  4. Table 1: and anything that does not pass those selections is a g-jet I guess? It would be good to be explicit. While it's true that this paper is to illustrate the method, the result will depend on the b-tagging efficiency, which is typically ~70% for the LHC experiments, quite a bit higher than what is illustrated here. How big of an impact does that have on the conclusions? (The choice of this tight b-tagging WP means the fake rate is low, which may make it hard to measure with the advertized precision: this becomes a few per-mille level measurement.

  5. Table 2: suggest to add the sqrt(s) and luminosity in the caption so people don't misinterpret the numbers.

  6. On p7 the absence of a large increase in the fitted nuisance parameters leads the authors to conclude your estimates are fine. But the fit uses pseudo-data as inputs, so the input likely doesn't contain any interesting systematic effects, so this is a bit of a circular argument. This conclusion therefore seems incorrect. Either the paper should estimate the impact of larger uncertainties (maybe hard to do), or just stick with the argument that these uncertainties are of the same order as those reported by the experiments.

  7. "Even using a suboptimal taggers, ...and without incorporating the full pT dependence... the model can capture...": but is it clear it would still be able to do that if the model were more complex and realistic? The input pseudo-data by definition do not deviate systematically from the MC used for the analysis...

  8. p12: the analysis result says 2.5 sigma, but the conclusions say 95% CL. This isn't quite the same thing. How is the 95% CL obtained?

Text:

  1. Bottom of p4 typo: The test statistics q => The test statistic q
  2. Fig 2 caption typo: showed => shown
  3. p 7 typos: statistics is => statistics are; lead => led
  4. p10 typo: is the statistics in (13), (14) => is the statistic in (13), (14)

  • validity: top
  • significance: high
  • originality: high
  • clarity: top
  • formatting: excellent
  • grammar: excellent

Author:  Darius Faroughy  on 2022-12-11  [id 3124]

(in reply to Report 1 on 2022-10-27)
Category:
answer to question

First of all, we agree with the general comment by the referee that statements in the text should not overestimate our certainty regarding the impact of incorporating more kinematic information in the analysis. We have modified the next to last paragraph of conclusions accordingly.

Regarding the requested changes:

  • 1) We agree with the referee that non-orthogonal taggers can be used, if they are combined in such a way that the resulting ${n_{b},n_{q}}$ bins are non-overlapping. The orthogonality we refer to in the text is of the resulting $b-$ and $q-$taggers as a whole, not of the specific algorithms which are combined to build these taggers. This is reflected in the strategy detailed in Section IV, as also noted by the referee, where we define the $q$-tagger as the intersection of the anti-$b_{2}$-tagger and the quark/gluon tagger.
  • 2) The referee makes a good point. We have added a sentence regarding the need for such a correction in the conclusions.
  • 3) We thank the referee for pointing out the mistake, we have corrected the notation.
  • 4) That is correct: any object that is neither a $b$ nor a $q$ is a $g$. We have added an additional sentence in caption of Table I for clarification. With respect to the specific choice of the benchmark, Section III is mainly a validation of the model with a choice of easy to understand (and to code) taggers. In Section IV we explore more common efficiencies such as a 70% WP and their impact on the method. In that sense, the specifics of taggers used in Section III are not very important as long as they are representative and allow us to trust the probabilistic model. This is specially true for the systematic uncertainties which are not obtained from thorough studies but by mimicking standard uncertainties and verifying that the resulting pulls are not too large.
  • 5) We thank the referee for the useful suggestion, we have added the information in the caption.
  • 6) In Section III we make use of simulations which are more complete than the probabilistic model implemented. We take the fact that the systematic uncertainties remain small as a confirmation that the model is able to capture the appropriate physics reasonably. We agree with the referee that because we have not included systematic variations of the data beyond those we hope to capture, it is not surprising the model works. Showing that this internal check (as well as the obtained consistency with CMS results) is validated is exactly the intention of the Section. If we expected other systematics to be relevant, we should include them in the model as well before running our tests in Section IV.
  • 7) The referee is certainly correct in pointing out that some effects cannot be known a priori before conducting dedicated studies. However, and as we mention in the last paragraph of the appendix and in the conclusions, incorporating $p_{T}$ can be achieved simply by partitioning the efficiency functions and adding the simulated $p_{T}$ distributions for each process. We have not done so mainly for computational reasons but there is no reason to think that the power of the analysis will decrease. Quite the opposite may well be true, adding more "categories" defined by the $p_{T}$ distribution should increase the statistical power of the analysis as the $p_{T}$ distributions are different for top and non-top jets. The added systematics should of course be included (mainly shape uncertainties) but the processes involved are fairly well understood and at low jet multiplicities the MC-uncertainties are fairly well under control.
  • 8) We meant to say ``above 95% CL" to reflect the fact that the discovery significance is above $2\sigma$. This is because the specific significance we obtain, which is higher, is obtained without considering the full set of systematics considered by the experimental collaborations, and we prefer to be conservative. We thank the referee for pointing out the inconsistency and we have corrected the text accordingly.
  • 9) We thank the referee for pointing these typos. We have corrected them in the manuscript.

---

## Round 1 · Referee Report · Anonymous (Referee 2) · 2022-10-31

Report

The authors propose an improved method to measure the CKM suppressed top decays t -> q W, with q = s,d at the LHC. Evidence for these decays has not been established yet. One motivation for the measurement of the top decays to light quarks is the direct determination of the CKM matrix elements V_ts and V_td. The standard methods to determine these CKM elements rely either on CKM unitarity or make use of loop processes which could be impacted by new physics.

Measurements of the top decays are challenging due to their small branching ratios and the difficulty of distinguishing them from the dominant t -> b W decay. The current most sensitive experimental analysis (ref. [3]) focuses on the ratio "R_b" of t -> b W decays to all top decays. The main observable that is used in this analysis is the number of b tagged jets in the event. The authors show that including also the number of light quark tagged jets can improve the sensitivity of the analysis by a factor of a few. They find that it might be possible to establish a value of R_b different from 1 at the high luminosity LHC.

In the paper, the authors describe in detail the sophisticated probabilistic model they use to extract R_b and how they have executed the analysis on Monte Carlo generated data. In fact, the paper is very technical and reads almost like an actual experimental analysis.

The results are encouraging and motivate continued efforts to measure top decays to light quarks at the LHC. I recommend publication, but would ask the authors to address a couple of minor points:

1) Figure 5 can be very useful in understanding the b tagging and light quark tagging procedures that the authors use. Unfortunately, it is not really explained what the Figure actually shows. I believe I was able to piece together that the different rings show the different steps of the procedure, and the unlabeled light brown area corresponds to the jets that were not b tagged in the first step. It would be great if the authors could add a detailed explanation of the Figure.

2) Figure 7 and 8 contain two lines labeled "not trustable". I do not understand what this refers to. The text explains that one of those lines is the maximal lepton-jet invariant mass from the top decay. What is the second line? What does "not trustable" refer to?

  • validity: -
  • significance: -
  • originality: -
  • clarity: -
  • formatting: -
  • grammar: -

Author:  Darius Faroughy  on 2022-12-11  [id 3123]

(in reply to Report 2 on 2022-10-31)

  • We thank the referee for pointing out the lack of clarity in Figure 5. We have added a detailed explanation in the caption.
  • Similarly, we thank the referee for pointing out the lack of clarity in Figures 7 and 8. We have modified them accordingly. The "not trustable" region, shown now hashed, is where Monte Carlo simulations do not yield trustable estimates. This, as detailed in Refs. [26] and [27], is because the region is populated by collinear lepton-jet pairs and we require leptons to be isolated, with jets separated from the selected leptons. We have modified the captions accordingly to include this definition.

---

## Round 1 · Referee Report · Anonymous (Referee 3) · 2022-11-1

Report

Authors address the question: Can the HL-LHC directly observe $V_{tq}$ CKM elements where $q = d,s$? In the Standard Model, those are precisely determined indirectly. A direct determination is yet another important test of the theory.

The starting point of the paper is the CMS analysis in Ref. [3]. The main idea is to additionally use $q$-taggers and the main result is that the HL-LHC will be able to establish non-zero off-diagonal CKM elements with top quark at more than 2 sigmas.

Before publication, I would like authors to address the following: - How is a constituent track, which counts toward multiplicity, defined? - Why not consider $p_T$-dependent multiplicity cut as suggested by Fig 7 of Ref. [21]? - How does pile-up impact this analysis? - What is the rationale behind the assumed systematic uncertainties in Table III? Very few details are provided regarding this point. Why not 3 times larger or smaller? What is the resulting change by a variation within a factor of 3?

  • validity: -
  • significance: -
  • originality: -
  • clarity: -
  • formatting: -
  • grammar: -

Author:  Darius Faroughy  on 2022-12-11  [id 3122]

(in reply to Report 3 on 2022-11-01)
Category:
answer to question

  • We thank the referee for pointing out the missing definition and have added it to the text. We refer to tracks as the reconstructed energy-flow tracks in the inner tracking volume that are simulated by Delphes.
  • We agree with the referee that considering a $p_{T}$ dependent multiplicity could enhance the performance of the tagger. However, the main goal of Section III was not to optimize the performance but rather to validate our model and obtain results that are consistent with the reported CMS analysis, we decided to use the simplest taggers possible. The choice we made is of course to some extent arbitrary and that is why in Section IV we implemented state-of-the-art taggers in order to optimize the performance.
  • We agree with the referee that the pile-out needs to be addressed, however, we believe that such study is beyond the scope of present work since our main goal was to stress the importance of including $n_q$ categories in a probabilistic model for $V_{tx}$ determination, while the analysis itself has several other limiting factors, with pile-up one of them. An in-detail analysis of pile-up mitigation effects, along with other experimental corrections, should of course be considered when deploying the proposed strategy to real data. We have added a comment along these lines in the conclusions.
  • The rationale was to consider systematics consistent with those reported by the experimental calibrations. The performance of our analysis is of course dependent on these systematics, with lower systematics increasing the statistical power. Given that the values of systematic uncertainties we assigned are reasonable (i.e. comparable to what is achieved in experiment), we expect that incorporating additional kinematical information to the probabilistic model will have a higher impact than modifying within reason the specific tagger systematic uncertainties.

---

## Editorial Decision

resubmitted